# MonoFlow: A Unified Generative Modeling Framework for Divergence GANs

## Abstract

Generative adversarial networks (GANs) play a minmax two-player game via adversarial training. The conventional understanding of adversarial training is that the discriminator is trained to estimate a divergence and the generator learns to minimize this divergence. We argue that despite the fact that many variants of GANs are developed following this paradigm, the existing theoretical understanding of GANs and the practical algorithms are inconsistent. In order to gain deeper theoretical insights and algorithmic inspiration for these GAN variants, we leverage Wasserstein gradient flows which characterize the evolution of particles in the sample space. Based on this, we introduce a unified generative modeling framework – MonoFlow: the particle evolution is rescaled via an arbitrary monotonically increasing mapping. Under our framework, adversarial training can be viewed as a procedure first obtaining MonoFlow's vector field via the discriminator and then the generator learns to parameterize the flow defined by the corresponding vector field. We also reveal the fundamental difference between variational divergence minimization and adversarial training. This analysis helps us to identify what types of generator loss functions can lead to the successful training of GANs and suggest that GANs may have more loss designs beyond those developed in the literature, e.g., non-saturated loss, as long as they realize MonoFlow. Consistent empirical studies are also included to validate the effectiveness of our framework.

## 1 Introduction

Generative adversarial nets (GANs) (Goodfellow et al., 2014; Jabbar et al., 2021) are a powerful generative modeling framework that has gained tremendous attention in recent years. GANs have achieved significant successes in applications, especially in high-dimensional image processing such as high-fidelity image generation (Brock et al., 2018; Karras et al., 2019), super-resolution (Ledig et al., 2017) and domain adaption (Zhang et al., 2017).

In the GAN framework, a discriminator $d$ and a generator $g$ play a minmax game. The discriminator is trained to distinguish real and fake samples and the generator is trained to generate fake samples to fool the discriminator. The equilibrium of the vanilla GAN is defined by[1]

$$\min_g \max_d V(g,d) = \mathbb{E}_{\mathbf{x} \sim p_{\text{data}}} \big\{ \log \sigma[d(\mathbf{x})] \big\} + \mathbb{E}_{\mathbf{z} \sim p_{\mathbf{z}}} \big\{ \log \big(1 - \sigma[d(g(\mathbf{z}))]\big) \big\} \tag{1}$$

The elementary optimization approach to solve the minmax game is adversarial training. Previous perspectives explained it as first estimating Jensen-Shannon divergence then the generator learns to minimize this divergence. Several variants of GANs have been developed based on this point of view for other probability divergences, e.g., $\chi^2$ divergence (Mao et al., 2017), Kullback-Leibler (KL) divergence (Arbel et al., 2021) and general $f$-divergences (Nowozin et al., 2016; Uehara et al., 2016), while others are developed with Integral Probability Metrics (Arjovsky et al., 2017; Dziugaite et al., 2015; Mroueh et al., 2018b). However, we emphasize that the traditional understanding over GANs is incomplete and here we present three non-negligible facts which are commonly associated with adversarial training, making it different from the standard variational divergence minimization (VDM) problem (Blei et al., 2017):

---

[1]We use a slightly different notation: $d(\mathbf{x})$ is the logit output of the classifier and $\sigma(\cdot)$ is the Sigmoid activation.

1. The estimated divergence is computed from the discriminator $d(\mathbf{x})$. It is a function only depending on samples $\mathbf{x}$ and cannot capture the variability of the generator's distribution $p_g$. However, the optimal discriminator in Proposition 1 by Goodfellow et al. (2014) requires $p_g$ to be a functional variable as well. This issue was also raised in (Metz et al., 2017; Franceschi et al., 2022).

2. The generator typically minimizes a divergence with a missing term, e.g., the vanilla GAN only minimizes the second term of Jensen-Shannon divergence $-\mathbb{E}_{\mathbf{z} \sim p_{\mathbf{z}}} \left\{ -\log \left( 1 - \sigma[d(g(\mathbf{z}))] \right) \right\}$ where $-\log \left( 1 - \sigma[d(g(\mathbf{z}))] \right)$ is a monotonically increasing function of $d(g(\mathbf{z}))$.

3. Practical algorithms are inconsistent with the theory, a heuristic trick "non-saturated loss" is commonly adopted to mitigate the gradient vanishing problem, but it still lacks a rigorous mathematical understanding. For example, the generator can minimize $-\mathbb{E}_{\mathbf{z} \sim p_{\mathbf{z}}} \left\{ \log \sigma[d(g(\mathbf{z}))] \right\}$ where $\log \sigma[d(g(\mathbf{z}))]$ is also a monotonically increasing function of $d(g(\mathbf{z}))$.

It is known the logit output $d(\mathbf{x})$ of a binary classification problem in Eq. (1) is the logarithm density ratio estimator between two distributions (Sugiyama et al., 2012). To gain a deeper understanding of adversarial training of GANs, we study the Wasserstein gradient flow of the KL divergence which characterizes a deterministic evolution of particles described by an ordinary differential equation (ODE). This ODE is a Euclidean gradient flow of a time-dependent log density ratio. Based on this ODE, we propose the **MonoFlow** framework – transforming the log density ratio by a monotonically increasing mapping such that the vector field of the gradient flow is rescaled along the same direction. Consequently, approximating and learning to parameterize MonoFlow is synonymous with adversarial training. Under our framework, we gain a comprehensive understanding of training dynamics over GANs: the discriminator obtains a bijection of the log density ratio that suggests the vector field and the generator learns to parameterize the particles of MonoFlow. All variants of divergence GANs are a subclass of our framework. Finally, we reveal that the discriminator and generator loss do not need to follow the same objective. The discriminator maximizes an objective to obtain a bijection of the log density ratio. Then the generator loss can be any monotonically increasing mapping of this log density ratio. Our contributions are summarized as follows:

- A novel generative modeling framework unifies divergence GANs, providing a new theoretically and practically consistent understanding of the underlying mechanism of the training dynamics over GANs.

- We reveal the fundamental difference between VDM and adversarial training, which indicates that the previous analysis of GANs based on the perspective of VDM might not provide benefits and instead we should treat GANs as a particle flow method.

- An analysis of what types of generator loss functions can lead to the success of training GAN. Our framework explains why and how non-saturated loss works.

- An algorithmic inspiration where GANs may have more variants of loss designs than we have already known.

## 2 WASSERSTEIN GRADIENT FLOWS

In this section, we review the definition of gradient flows in Wasserstein space $(\mathcal{P}(\mathbb{R}^n), W_2)$, the space of Borel probability measures $\mathcal{P}(\mathbb{R}^n)$ defined on $\mathbb{R}^n$ with finite second moments and Eq.uipped with the Wasserstein-2 metric. An absolutely continuous curve of probability measures $\{q_t\}_{t \geq 0} \in \mathcal{P}(\mathbb{R}^n)$ is a Wasserstein gradient flow if it satisfies the following continuity equation (Ambrosio et al., 2008),

$$\frac{\partial q_t}{\partial t} = \operatorname{div}\left(q_t \nabla_{W_2} \mathcal{F}(q_t)\right), \tag{2}$$

where $\nabla_{W_2} \mathcal{F}(q_t)$ is called the Wasserstein gradient of the functional $\mathcal{F} : \mathcal{P}(\mathbb{R}^n) \to \mathbb{R}$. The Wasserstein gradient is defined as $\nabla_{\mathbf{x}} \frac{\delta \mathcal{F}}{\delta q_t}$, i.e. the Euclidean gradient of the functional's first variation $\frac{\delta \mathcal{F}(q_t)}{\delta q_t}$. Specifically, for the functional $\mathcal{F}(q_t) = \int \log \frac{q_t}{p} dq_t$ as the KL divergence where $p$ is a fixed target probability measure, we have $\frac{\delta \mathcal{F}(q_t)}{\delta q_t} = \log \frac{q_t}{p} + 1$. Hence, the Wasserstein gradient flow of

Figure 1: The illustration of a Wasserstein gradient flow and its particle evolution. In Wasserstein space, the blue curve is a gradient flow and the red dotted line is a geodesic. $q_t$ evolves along a curve whose tangent vector is given by $-\nabla_{W_2}\mathcal{F}(q_t)$ such that the functional is always decreasing with time. Correspondingly, particles evolve in Euclidean space towards the target measure $p$ with the vector field $-\nabla_{\mathbf{x}}\frac{\delta\mathcal{F}}{\delta q_t}(\mathbf{x})$. Note that directly minimizing the Wasserstein-2 metric $W_2(q_t, p)$ instead yields a path $\{q_t\}_{t\geq 0}$ along the geodesic connecting $q_0$ and $p$ over Wasserstein space.

the KL divergence reads the Fokker-Planck equation,

$$\frac{\partial q_t}{\partial t} = \mathrm{div}\big(q_t(\nabla_{\mathbf{x}}\log q_t - \nabla_{\mathbf{x}}\log p)\big). \tag{3}$$

As $t \to \infty$, the stationary probability measure of $q_t$ is the target $p$. Denote the Euclidean path of random variables as $\{\mathbf{x}_t\}_{t\geq 0} \in \mathbb{R}^n$ with the initial condition $\mathbf{x}_0 \sim q_0$, we can define an ordinary differential equation (ODE) which describes the evolution of particles in $\mathbb{R}^n$,

$$\mathrm{d}\mathbf{x}_t = \big(\nabla_{\mathbf{x}}\log p(\mathbf{x}_t) - \nabla_{\mathbf{x}}\log q_t(\mathbf{x}_t)\big)\mathrm{d}t := v_t(\mathbf{x}_t)\mathrm{d}t, \quad \mathbf{x}_0 \sim q_0, \tag{4}$$

where the vector field $v_t$ of these particles is the negative Euclidean gradient of the functional's first variation. As shown in Figure 1, Wasserstein gradient flows establish a connection between the probability evolution in Wasserstein space and its associated particle evolution in Euclidean space.

Applying Itô integral to Langevin dynamics $\mathrm{d}\mathbf{x}_t = \nabla_{\mathbf{x}}\log p(\mathbf{x}_t)\mathrm{d}t + \sqrt{2}\mathrm{d}\mathbf{w}$ where $\mathrm{d}\mathbf{w}$ is a Wiener process, we obtain the same Fokker-Planck equation in Eq. (3). This indicates that the deterministic particle evolution by the ODE can be approximated via a stochastic differential equation (SDE). Langevin dynamics admits the same marginal probability measure $q_t$ as Eq. (4), this relation of SDE and its corresponding ODE were also studied in score-based diffusion models (Song et al., 2021). Langevin dynamics was first interpreted as the Wasserstein gradient flow of the KL divergence by Jordan et al. (1998); Otto (2001). It plays an important role in generative modeling as a sampling scheme. In order to transform noises into the target data distribution by Langevin dynamics, an essential step is to fit the data distribution using energy-based models (Song & Kingma, 2021) or to directly estimate its scores with score-matching techniques (Hyvärinen & Dayan, 2005; Vincent, 2011; Song & Ermon, 2019).

## 3 MonoFlow: A Unified Generative Modeling Framework

This section presents our main contribution that connects gradient flows and divergence GANs into a unified framework. We first introduce MonoFlow where the ODE evolution is rescaled via a monotonically increasing function. Consequentially, learning to parameterize the rescaled flow by a neural network recovers the bi-level optimization dynamics of training GANs. This gives us a novel understanding of the hidden mechanism of adversarial training.

### 3.1 MonoFlow

We consider the ODE in Eq. (4) with a fixed target measure $p$, e.g., a data distribution in a generative modeling scenario. Assume that we have a time-dependent log density ratio function as $\log r_t(\mathbf{x}) = \log\frac{p(\mathbf{x})}{q_t(\mathbf{x})}$, the ODE can be rewritten as

$$\mathrm{d}\mathbf{x}_t = \nabla_{\mathbf{x}}\log r_t(\mathbf{x}_t)\mathrm{d}t, \quad \mathbf{x}_0 \sim q_0. \tag{5}$$

This is a gradient flow in Euclidean space where its vector field is the gradient of the log density ratio. With a monotonically increasing (strict) mapping $h\colon \mathbb{R} \to \mathbb{R}$ where $h$ is first-order differentiable, we can define another ODE:

$$\mathrm{d}\mathbf{x}_t = \nabla_{\mathbf{x}}h\big(\log r_t(\mathbf{x}_t)\big)\mathrm{d}t = h'\big(\log r_t(\mathbf{x}_t)\big)\nabla_{\mathbf{x}}\log r_t(\mathbf{x}_t)\mathrm{d}t, \quad \mathbf{x}_0 \sim q_0 \tag{6}$$

By transforming the time-dependent log density ratio under the mapping $h$, its first-order derivative rescales the vector field of the original gradient flows defined in Eq. (5). We call Eq. (6) as **MonoFlow**.

MonoFlow defines a different family of vector fields $\{v_t\}_{t \geq 0}$ for the particle evolution where $v_t(\mathbf{x}_t) = h'\big(\log r_t(\mathbf{x}_t)\big)\nabla_{\mathbf{x}} \log r_t(\mathbf{x}_t)$. Conversely, the vector fields $\{v_t\}_{t \geq 0}$ also determine an absolutely continuous curve $\{q_t\}_{t \geq 0}$ in Wasserstein space by the continuity equation, see Theorem 4.6 in (Ambrosio et al., 2008),

$$\frac{\partial q_t}{\partial t} = -\text{div}(q_t v_t), \tag{7}$$

under mild regularity conditions. Hence the probability evolution of MonoFlow is described by

$$\frac{\partial q_t}{\partial t} = \text{div}\big(D_t \nabla_{\mathbf{x}} q_t\big) - \text{div}\big(\zeta_t^{-1} q_t \nabla_{\mathbf{x}} \log p\big), \tag{8}$$

where $D_t = \zeta_t^{-1} = h'(\log r_t)$. Eq. (8) is a special case of convection-diffusion equations where $D_t$ is called the diffusion coefficient and $\zeta_t^{-1}$ is called mobility. MonoFlow defines a positive diffusion coefficient. This has a physical interpretation that particles diffuse to spread probability mass over the target measure other than concentrate. Next, we study the properties of MonoFlow.

**Theorem 3.1.** *If $h$ is strictly increasing, i.e., $h'(\cdot) > 0$, the dissipation rate $\frac{\partial \mathcal{F}(q_t)}{\partial t}$ for the KL divergence $\mathcal{F}(q_t) = \int \log \frac{q_t}{p} dq_t$ satisfies*

$$\frac{\partial \mathcal{F}(q_t)}{\partial t} \leq 0, \tag{9}$$

*the equality is achieved if and only if $q_t = p$ and the marginal probability $q_t$ of MonoFlow evolves to $p$ as $t \to \infty$.*

Theorem 3.1 shows that MonoFlow does not disturb the equilibrium of Eq. (3), and the convergence to the target probability measure $p$ is guaranteed. The negative dissipation rate ensures that the gradient flow curve $\{q_t\}_{t \geq}$ of MonoFlow always decreases the KL divergence with time.

MonoFlow is obtained by transforming the log density ratio which arises from the Wasserstein gradient flow of KL divergence. We can also obtain other forms of deterministic particle evolution by considering Wasserstein gradient flows of general $f$-divergences,

$$\mathcal{D}_f(p\|q) = \int f\left(\frac{p}{q}\right) dq, \tag{10}$$

where $f : \mathbb{R}^+ \to \mathbb{R}$ is a convex function and $f(1) = 0$.

**Theorem 3.2.** *The Wasserstein gradient flow of an $f$-divergence characterizes the evolution of particles in $\mathbb{R}^n$ by*

$$d\mathbf{x}_t = r(\mathbf{x}_t)^2 f''\big(r(\mathbf{x}_t)\big)\nabla_{\mathbf{x}} \log r_t(\mathbf{x}_t) dt, \quad \mathbf{x}_0 \sim q_0. \tag{11}$$

A similar result can also be derived with the reversed $f$-divergences $\mathcal{D}_f(q\|p)$ used by (Gao et al., 2019; Ansari et al., 2021). Theorem 3.2 shows that the particle evolution of the Wasserstein gradient flow of $f$-divergences is a special instance of MonoFlow where $h'(\log r) = r^2 f''(r) > 0$ because $f$ is a convex function. It indicates once a curve $\{q_t\}_{t \geq 0}$ evolves with the time $t$ in Wasserstein space to decrease an $f$-divergence, it simultaneously decreases the KL divergence as well since the dissipation rate of MonoFlow is negative.

## 3.2 Practical Approximations of Density Ratios

We first discretize the ODE in Eq. (6) by the forward Euler method such that we obtain standard gradient ascent iterations with step size $\alpha$ and the index of the discretized time step $k$ [2]:

$$\mathbf{x}_{k+1} = \mathbf{x}_k + \alpha \nabla_{\mathbf{x}} h\big(\log r_k(\mathbf{x}_k)\big), \quad t_{k+1} = t_k + \alpha. \tag{12}$$

Therefore, we can sample initial particles $\mathbf{x}_0 \sim q_0$ and perform gradient ascent iterations by estimating the density ratio $r_k(\mathbf{x}) = \frac{p(\mathbf{x})}{q_k(\mathbf{x})}$ using samples from $q_k$ and $p$. In order to enable a practical

---

[2]For the sake of simplicity, we briefly replace $t_k$ by its index $k$, though it is not rigorous.

Table 1: Different types of divergence GANs. $f$ is a convex function and $\tilde{f}$ is the convex conjugate by $\tilde{f}(d) = \sup_{r \in \mathrm{dom} f}\{rd - f(r)\}$.

| | $\phi(d)$ | $\psi(d)$ | $d^*(\mathbf{x})$ | $h_{\mathcal{T}}(d)$ |
|---|---|---|---|---|
| Vanilla GAN | $\log \sigma(d)$ | $\log(1 - \sigma(d))$ | $\log r(\mathbf{x})$ | $-\log(1 - \sigma(d))$ |
| Non-saturated GAN | $\log \sigma(d)$ | $\log(1 - \sigma(d))$ | $\log r(\mathbf{x})$ | $\log \sigma(d)$ |
| $f$-GAN | $d$ | $-\tilde{f}(d)$ | $f'(r(\mathbf{x}))$ | $d$ |
| $b$-GAN | $f'(d)$ | $f(d) - df'(d)$ | $r(\mathbf{x})$ | $df'(d) - f(d)$ |
| Least-square GAN | $-(d-1)^2$ | $-d^2$ | $\frac{r(\mathbf{x})}{1 + r(\mathbf{x})}$ | $-(d-1)^2$ |
| Generalized EBM (KL) | $-(d + \lambda)$ | $-\exp(-d - \lambda)$ | $-\log r(\mathbf{x}) - \lambda$ | $\exp(-d - \lambda)$ |

algorithm to obtain the time-dependent density ratio, we introduce a general framework that solves the following optimization problem, similar to Moustakides & Basioti (2019),

$$\max_{d \in \mathcal{H}} \left\{ \mathbb{E}_{\mathbf{x} \sim p}\left[\phi\big(d(\mathbf{x})\big)\right] + \mathbb{E}_{\mathbf{x} \sim q_k}\left[\psi\big(d(\mathbf{x})\big)\right] \right\}, \tag{13}$$

where $d \colon \mathbb{R}^n \to \mathbb{R}$ is a discriminator and $\mathcal{H}$ is a class of measurable functions. $\phi$ and $\psi$ are scalar functions upon design later.

**Lemma 3.3.** *Solving Eq. (13), the optimal $d^*$ satisfies*

$$d^*(\mathbf{x}) = \mathcal{T}^{-1}(r(\mathbf{x})), \quad r(\mathbf{x}) = \frac{p(\mathbf{x})}{q_k(\mathbf{x})}, \tag{14}$$

*with $\mathcal{T}(d(\mathbf{x})) := -\frac{\psi'(d(\mathbf{x}))}{\phi'(d(\mathbf{x}))}$. Note that the resulting mapping $\mathcal{T}$ must be a bijection such that its inverse exists. Additional conditions on the hypothesis of $\phi$ and $\psi$ can be found in Appendix A.3.*

**Remark:** Note that two-sample density ratio estimations discard the density information from $q_k$. The functions $d(\mathbf{x})$, $r(\mathbf{x})$ only depend on $\mathbf{x}$ and they cannot capture the variability of $q_k$.

To this end, we can train $d$ to solve the optimization problem in Eq. (13) and the density ratio is approximated by $\mathcal{T}(d(\mathbf{x}))$. For example, in a standard binary classification problem where we can design $\phi(d) = \log \sigma(d)$ and $\psi(d) = \log(1 - \sigma(d))$, we have $d^*(\mathbf{x}) = \log r(\mathbf{x})$ (Sugiyama et al., 2012). Other types of density ratio estimation can be found in Table 1 as they have been already used in GAN variants where $q_k$ refers to the generator's distribution $p_g$. In practice, since the change of $\mathbf{x}_k$ is sufficiently small at every step $k$, we can use a single discriminator $d(\mathbf{x})$ and perform a few gradient updates to solve Eq. (13) per iteration $k$ to approximate the time-dependent density ratio $r_k(\mathbf{x})$, which is identical to GAN training.

### 3.3 Parameterization of the Discretized MonoFlow

The previous method directly pushes particles in the Euclidean space towards the target distribution. We can use a neural network generator to mimic the distribution of these particles, i.e., the generator learns to draw samples. If we parameterize particles with a neural network generator $g_\theta$ that takes as input random noises $\mathbf{z} \sim p_{\mathbf{z}}$ and output particles $\mathbf{x}_{(\theta, \mathbf{z})} = g_\theta(\mathbf{z})$, the infinitesimal change of parameters of the generator is

$$\frac{\mathrm{d}\theta_t}{\mathrm{d}t} = \int \frac{\partial g_{\theta_t}(\mathbf{z})}{\partial \theta_t} \nabla_{\mathbf{x}} h\big(\log r_t(\mathbf{x}_t)\big) p_{\mathbf{z}}(\mathbf{z}) \mathrm{d}\mathbf{z}, \quad \text{where } \mathbf{x}_t = g_{\theta_t}(\mathbf{z}), \tag{15}$$

where $\frac{\partial g_{\theta_t}(\mathbf{z})}{\partial \theta_t}$ is the Jacobian of the neural network generator. Apply the forward Euler method with the step size $\beta$, we have

$$\theta_{k+1} = \theta_k + \beta \nabla_\theta \mathbb{E}_{\mathbf{z} \sim p_{\mathbf{z}}}\left[h\big(\log r_k(g_{\theta_k}(\mathbf{z}))\big)\right]. \tag{16}$$

Eq. (15) can be regarded as amortizing particles of the gradient flow into a neural network generator by approximately solving $\theta_{k+1} = \arg\min_\theta \mathbb{E}_{\mathbf{z} \sim p_{\mathbf{z}}}\|g_\theta(\mathbf{z}) - \mathbf{x}_{k+1}\|^2$ where $\mathbf{x}_{k+1} = g_{\theta_k}(\mathbf{z}) + \alpha \nabla_{\mathbf{x}} h\big(\log r_k(g_{\theta_k}(\mathbf{z}))\big)$ with a one-step gradient descent (Wang & Liu, 2017).

Consequently, by the chain rule we have $\mathrm{d}\mathbf{x}_t = \frac{\partial g_{\theta_t}(\mathbf{z})}{\partial \theta_t} \mathrm{d}\theta_t$, replace $\mathrm{d}\theta_t$ with Eq. (15),

$$\mathrm{d}\mathbf{x}_t = \mathbb{E}_{\mathbf{z}' \sim p_{\mathbf{z}}}\left[K_g^t(\mathbf{z}, \mathbf{z}') \nabla_{\mathbf{x}} h\big(\log r_t(\mathbf{x}_t)\big)\right] \mathrm{d}t \tag{17}$$

where $K_g^t(\mathbf{z}, \mathbf{z}') = \langle \frac{\partial g_{\theta_t}(\mathbf{z})}{\partial \theta_t}, \frac{\partial g_{\theta_t}(\mathbf{z}')}{\partial \theta_t} \rangle$ is the neural tangent kernel (NTK) (Jacot et al., 2018) defined by the generator. Note that Eq. (17) realizes Stein Variational Gradient Descent (Liu & Wang, 2016; Franceschi et al., 2022) if $h$ is an identity mapping.

## 3.4 A UNIFIED FORMULATION OF DIVERGENCE GENERATIVE ADVERSARIAL NETS

Based on the above derivation, we propose a general formulation for divergence GANs. We clarify that GANs can be treated with different objective functions for training discriminators and generators. All of these variants are algorithmic instantiations of the parameterized MonoFlow. The unified framework is summarized as: given a discriminator $d$ and a generator $g$, the discriminator $d$ learns to maximize

$$\mathbb{E}_{\mathbf{x} \sim p_{\text{data}}} \left[ \phi\big(d(\mathbf{x})\big) \right] + \mathbb{E}_{\mathbf{z} \sim p_{\mathbf{z}}} \left[ \psi\big(d(g(\mathbf{z}))\big) \right], \tag{18}$$

where $p_{\text{data}}$ refers to the data distribution. Next, we train the generator $g$ to minimize

$$-\mathbb{E}_{\mathbf{z} \sim p_{\mathbf{z}}} \left[ h_{\mathcal{T}}\big(d(g(\mathbf{z}))\big) \right]. \tag{19}$$

where $h_{\mathcal{T}}(d) = h(\log \mathcal{T}(d))$ and $h$ can be any strictly increasing function. We summarize some typical GAN variants in Table 1. We view adversarial training as maximizing Eq. (18) to obtain the density ratio which suggests the vector field for MonoFlow and minimizing Eq. (19) as learning to parameterize MonoFlow corresponding to Eq. (16). Especially, we find that $b$-GAN (Uehara et al., 2016) realizes parameterized Wasserstein gradient flows of $f$-divergences since its generator loss is aligned with Theorem 3.2 for the design of $h$.

## 4 UNDERSTANDING ADVERSARIAL TRAINING VIA MONOFLOW

The dominating understanding of adversarial training over GANs is that the generator learns to minimize the divergence estimated from the discriminator. However, as pointed out in Section 1, the theoretical explanation of GAN and its practical algorithm are inconsistent. In this section, through the lens of MonoFlow, we will explain why this inconsistency of adversarial training still can lead to convergence to the target distribution and how it differs from a variational divergence minimization (VDM) problem for generative modeling.

### 4.1 WHY THE ADVERSARIAL GAME WORKS?

In an adversarial game, the discriminator is trained to maximize the lower bound of $f$-divergences. This lower bound can be derived via the dual representation of $f$-divergences (Nguyen et al., 2010) between $p_{\text{data}}$ and $p_g$,

$$\mathcal{D}_f(p_{\text{data}}||p_g) = \max_{d \in \mathcal{H}} \Big\{ \underbrace{\mathbb{E}_{\mathbf{x} \sim p_{\text{data}}} \big[ d(\mathbf{x}) \big] - \mathbb{E}_{\mathbf{x} \sim p_g} \big[ \tilde{f}\big(d(\mathbf{x})\big) \big]}_{\text{lower bound}} \Big\}, \quad r(\mathbf{x}) = \frac{p_{\text{data}}(\mathbf{x})}{p_g(\mathbf{x})}, \tag{20}$$

where $\tilde{f}(d) = \sup_{r \in \text{dom} f} \{rd - f(r)\}$ is the convex conjugate of $f(r)$ and $\mathcal{H}$ is a class of any measurable functions. Note that for binary classification problems where we design specific $\phi$ and $\psi$, the corresponding optimization problem in Eq. (18) can be translated into an equivalent formulation as the above dual representation (Nowozin et al., 2016). Since the first term of the lower bound in Eq. (20) is irrelevant to $p_g$, the generator actually only learns to minimize the second term,

$$\min_g -\mathbb{E}_{\mathbf{x} \sim p_g} \big[ \tilde{f}\big(d(\mathbf{x})\big) \big] \tag{21}$$

Meanwhile, the generator can also alternatively minimize the heuristic non-saturated loss $-\mathbb{E}_{\mathbf{x} \sim p_g} \big[ d(\mathbf{x}) \big]$, which has been proven to work well in practice. By the Fenchel duality, the optimal $d^*$ is given by $d^* = f'(r)$ with the equality $\tilde{f}(d^*) = rf'(r) - f(r)$. Fortunately, it can be simply verified that $f'(r)$ and $rf'(r) - f(r)$ are both monotonically increasing functions of the density ratio (as well as the log density ratio). Hence, adversarial training with the vanilla loss and the non-saturated loss both fall into the framework of MonoFlow which has theoretical guarantees.

Table 2: Comparisons the convergence when using different $f$ and $h$ on three density ratio models: "✓" means the generator converges to the data distribution and "✗" means it does not converge. For $r(\mathbf{x}, \theta)$ and $r(\mathbf{x}, \theta_{\text{de}})$, the convergences means the generator parameter $\theta$ can converge to the true value. For $r_{\text{GAN}}(\mathbf{x})$, convergence means the parameter can closely approximate the true value. Visualization results are included in Appendix B.1.

|  | if $f$ convex | if $h$ mono increases | $r(\mathbf{x}, \theta)$ | $r(\mathbf{x}, \theta_{\text{de}})$ | $r_{\text{GAN}}(\mathbf{x})$ |
|---|---|---|---|---|---|
| KL | Yes | Yes | ✓ | ✓ | ✓ |
| Forward KL | Yes | No | ✓ | ✗ | ✗ |
| Chi-Square | Yes | No | ✓ | ✗ | ✗ |
| Hellinger | Yes | No | ✓ | ✗ | ✗ |
| Jensen-Shannon | Yes | No | ✓ | ✗ | ✗ |
| Exp | No | Yes | ✗ | ✓ | ✓ |

## 4.2 DIFFERENCE BETWEEN ADVERSARIAL TRAINING AND VARIATIONAL DIVERGENCE MINIMIZATION

VDM differs from adversarial training because it requires density information from the generator, as elaborated in the following. In variational divergence minimization, the generator $g_\theta$ usually defines a distribution via an explicit density function $p_g(\mathbf{x}; \theta)$. For example, $\mathbf{x}$ can be reparameterized as a Gaussian random variable where $\theta$ are its mean and scale. We are interested in minimizing an $f$-divergence

$$\mathcal{D}_f(p_{\text{data}} \| p_g) = \mathbb{E}_{\mathbf{x} \sim p_g}\big[f\big(r(\mathbf{x}, \theta)\big)\big] = \text{Cost}(\theta), \qquad (22)$$

where the density ratio $r(\mathbf{x}, \theta) = \frac{p_{\text{data}}(\mathbf{x})}{p_g(\mathbf{x}; \theta)}$ is a function of $\mathbf{x}$ as well as $\theta$ to capture the variability of $p_g$. Integrate out $\mathbf{x}$, the $f$-divergence can be written as a cost function of $\theta$. Since $f$ is convex, such that by Jensen inequality, this cost is minimized at zero when $r(\mathbf{x}, \theta)$ is a constant for each $\mathbf{x}$, meaning $p_g = p_{\text{data}}$. Similarly under Fenchel-duality, we can rewrite $f$-divergences as

$$\text{Cost}(\theta) = \mathbb{E}_{\mathbf{x} \sim p_{\text{data}}}\big[d^*(\mathbf{x}, \theta)\big] - \mathbb{E}_{\mathbf{x} \sim p_g}\big[\tilde{f}\big(d^*(\mathbf{x}, \theta)\big)\big], \qquad (23)$$

where $d^*(\mathbf{x}, \theta) = f'(r(\mathbf{x}, \theta))$. Eq. (23) is different from Eq. (20), where $\text{Cost}(\theta)$ depends on the first term of the dual representation. However, in an adversarial training scenario, the density ratio estimator $r(\mathbf{x})$ or its bijection $d(\mathbf{x})$ are only functions of the sample $\mathbf{x}$. It was also discussed by Metz et al. (2017); Franceschi et al. (2022), the smoothness between the discriminator and the variability of $p_g$ is lost during the practical algorithm. Plugging $r(\mathbf{x})$ or $d(\mathbf{x})$ into the $f$-divergence to replace $r(\mathbf{x}, \theta)$ or $d^*(\mathbf{x}, \theta)$, we can approximate $f$-divergences but the approximated divergences can never be viewed as a cost function of $\theta$ anymore. This is the major disconnection between the theory and the practical algorithm over GANs, as Eq. (4) in (Goodfellow et al., 2014) is similar to Eq. (23).

## 4.3 EMPIRICAL STUDY OF GAUSSIANS

This part highlights the differences between density ratio models in terms of the convergence of generators. We start from the simplest form of a generator $p_{g_\theta} : \mathbf{x} = \mu + s\mathbf{z}$, where $\mathbf{z} \sim N(0, I)$ and $\theta = (\mu, s)$, where $\mu$ is the mean and $s$ is the scale matrix. Let the data distribution be $p_{\text{data}} = N(\mu_0, s_0^T s_0)$. By assuming the generator and data distributions are Gaussians, we can define three density ratio models. The first model is $r(\mathbf{x}, \theta) = \frac{p_{\text{data}}(\mathbf{x})}{p_g(\mathbf{x}; \theta)}$, where the density ratio function depends on $\mathbf{x}$ and $\theta$ simultaneously. The second model is $r(\mathbf{x}, \theta_{\text{de}}) = \frac{p_{\text{data}}(\mathbf{x})}{p_g(\mathbf{x}; \theta_{\text{de}})}$, where $\theta_{\text{de}}$ means we detach the gradient of $\theta$ such that the second model cannot reflect the variability of $p_g$. The third model is $r_{\text{GAN}}(\mathbf{x})$ where the density ratio is approximated by performing a single gradient update for the classifier using samples in standard adversarial training. We train the generator to minimize the following loss function with the above three density ratio models respectively (for $r_{\text{GAN}}(\mathbf{x})$, we use the standard bi-level training),

$$\min_\theta \mathbb{E}_{\mathbf{z} \sim p_\mathbf{z}}\big[f(r)\big] \text{ or equivalently } \min_\theta -\mathbb{E}_{\mathbf{z} \sim p_\mathbf{z}}\big[h(\log r)\big] \qquad (24)$$

Given $f(r)$ we can rewrite it as a function of log density ratio $h(\log r) = -f(r)$. Similarly, given $h(\log r)$, we can write it as a function of density ratio $f(r) = -h(\log r)$. In this experiment, we consider five types of $f$-divergences (expressions summarized in Appendix B.1). In addition, we study a strictly increasing function $h$ given by $h(\log r) = \exp\big(1.5 \log r\big) = r^{1.5}$ where its

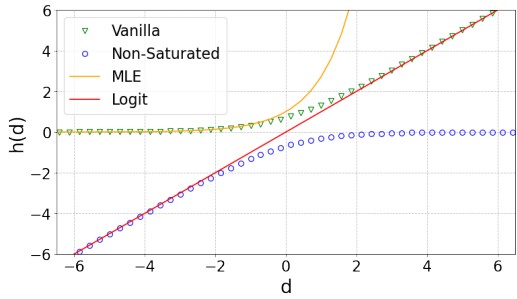 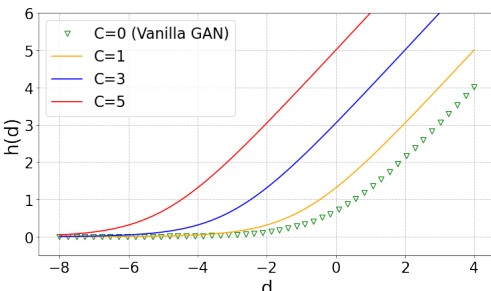

Figure 2: The plot of different generator losses as a function of $d$.

Figure 3: The plots of the vanilla losses by adding different $C$s.

$f(r) = -r^{1.5}$ is concave. The results are summarized in Table 2, which verifies our analysis that the objective of VDM should be a convex function of the density ratio, whereas MonoFlow only requires it to be an increasing function of the log density ratio. $r(\mathbf{x}, \theta_{\text{de}})$ can recover the true $f$-divergence, but minimizing this $f$-divergence has no effects except for KL divergence.

**Remark**: Minimizing the KL divergence using the detached model $r(\mathbf{x}, \theta_{\text{de}})$ also leads to convergence. This mechanism is called "Variational Inference via Wasserstein Gradient Flows" (Lambert et al., 2022) and it also have a lower variance (Roeder et al., 2017).

## 5 ALGORITHIMIC INSIGHTS: ALTERNATIVES OF GENERATOR LOSS

### 5.1 ANALYZING PRACTICAL EFFECTIVENESS OF GENERATOR LOSS

In this part, we explain what types of generator loss functions can lead to the successful training of GANs via the lens of MonoFlow. We provide a study for the vanilla GAN since the logit output of the binary classifier is the log density ratio where we have $d(\mathbf{x}) = \log r(\mathbf{x})$ (see Table 1). We consider four generator losses which are monotonically increasing functions of the log density ratio: **1). Vanilla loss**: $h(d) = -\log(1 - \sigma(d))$; **2). Non-saturated (NS) loss**: $h(d) = \log(\sigma(d))$. **3). Maximum likelihood loss**: $h(d) = \exp(d)$. **4). Logit loss**: $h(d) = d$.

The plot of these functions is shown in Figure 2. The vanilla loss and the MLE loss do not work well in practice (Goodfellow, 2016), since at initial training steps, the generator is weak and $d(\mathbf{x}) = \log \frac{p_{\text{data}}(\mathbf{x})}{p_g(\mathbf{x})} \ll 0$, for $\mathbf{x} \sim p_g(\mathbf{x})$. We may observe in Figure 2, the curves of the vanilla loss and the MLE loss are fairly flat on the left side, which means the derivative $h'(\cdot)$ is nearly zero. According to Eq. (6), such a rescaling scheme yields extremely small vector fields, resulting in the generator being trapped at the initial steps as the infinitesimal change of particles $d\mathbf{x}_t \approx 0$. The NS and logit loss both have non-zero derivatives when $d(\mathbf{x}) < 0$ despite that the NS loss is flat on the right side. This is not a problem since $d(\mathbf{x})$ gradually increases from a negative value during the training and when $d(\mathbf{x}) = 0$, it means $p_g = p_{\text{data}}$ such that the generator converges.

### 5.2 AN EMBARRASSINGLY SIMPLE TRICK TO FIX VANILLA GAN

We have justified that GANs can work with any generator loss as long as it is a monotonically increasing mapping of the log density ratio and this mapping's derivative deviates from zero when the $\log r(\mathbf{x}) \ll 0$. We show the effects of shifting the generator loss of the vanilla GAN left by adding a constant $C$ to the sigmoid function,

$$h(d) = -\log(1 - \sigma(d + C)) \tag{25}$$

By adding a constant, we can obtain a better monotonically increasing function whose derivative deviates from zero significantly, see Figure 3. The neural network architecture used here is DCGAN (Radford et al., 2015) and we follow the vanilla GAN framework where the log density ratio is obtained by logit output from the binary classifier and the model is trained with 15 epochs. The generated samples are shown in Figure 4. We observe that when $C = 3$ and $C = 5$, the generator losses in Eq. (25) begin to work, i.e., generators output plausible fake images.

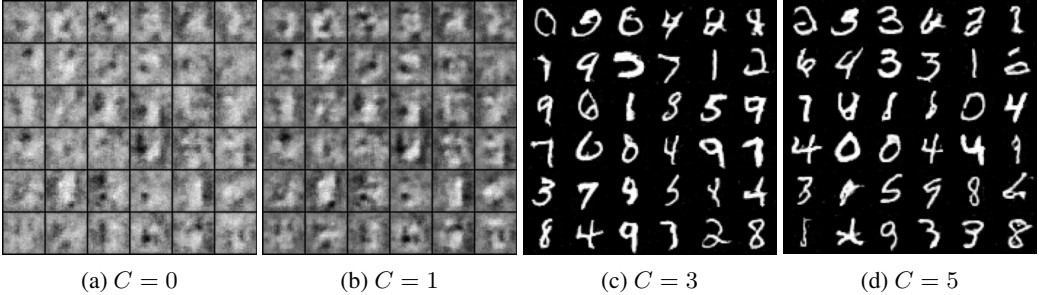

| (a) $C = 0$ | (b) $C = 1$ | (c) $C = 3$ | (d) $C = 5$ |

Figure 4: Generated samples with different $C$s. Experiments uses the data set MNIST.

## 6 RELATED WORKS

**Gradient Flow**: Wasserstein gradient flows of $f$-divergences have been previously studied in deep generative modeling as a refinement approach to improve sample quality (Ansari et al., 2021). A close work to ours is (Gao et al., 2019) where the authors proposed to use gradient flows of $f$-divergences to refine fake samples output by the generator and the generator learns to minimize the squared distance between the refined samples and the original fake samples. However, neither of the above reveals the equivalence between gradient flows and divergence GANs. Furthermore, MonoFlow is a more generalized framework to cover existing gradient flows of $f$-divergences and our method also applies to traditional loss designs as well as many other types of monotonically increasing functions. **IPM GANs**: Our framework unifies divergence GANs since estimating a probability divergence is naturally related to density ratio estimation (Sugiyama et al., 2012). However, some variants of GANs are developed with Integral Probability Metric (IPM) (Sriperumbudur et al., 2009). For example, WGANs (Arjovsky et al., 2017; Gulrajani et al., 2017) estimate the Wasserstein-1 metric and then minimize this metric. While MonoFlow is associated with Wasserstein-2 metric, minimizing a functional in $\mathcal{P}(\mathbb{R}^d)$ naturally decreases Wasserstein-2 metric as well. Other types of IPM GANs are MMD GAN (Dziugaite et al., 2015) and Sobolev GAN (Mroueh et al., 2018b). Both of them have been interpreted as gradient flow approaches (Mroueh & Nguyen, 2021; Mroueh et al., 2018a) but associated with different vector fields. Franceschi et al. (2022) studied the NTK view on GANs given a vector field specified by a loss function of IPM but lacks connections to divergence GANs. **Diffusion Models**: diffusion models (Ho et al., 2020; Song et al., 2021; Luo, 2022) are another line of generative modeling framework. This framework first perturbs data by adding noises with different scales to create a path $\{q_t\}_{t \geq 0}$ interpolating the data distribution and the noise distribution. Subsequently, the generative modeling is to reverse $\{q_t\}_{t \geq 0}$ as denoising. The similarity of MonoFlow and diffusion models is that they both involves particle evolution associated with different paths of marginal probabilities. The difference is as follows: the vector field of MonoFlow is obtained with the log density ratio and the log density ratio must be corrected per iteration by gradient update; diffusion models directly estimate vector fields by time-dependent neural networks and they are straightforward particle methods.

## 7 CONCLUSIONS

We introduce a unified framework for GAN variants. Our framework provides a comprehensive understanding to help us get insights into why and how adversarial training works. The mechanism of adversarial training is not as adversarial as we used to think. It instead simulates an ODE system, the bi-level step of adversarial can be regarded as it first estimates the vector field of MonoFLow and next the generator is updated to learn to draw particles guided by the vector field, aka we call it parameterizing MonoFlow. Therefore, all GAN variants discussed in this paper are equal at the methodology level. They all are different methods of estimating the bijection of the log density ratio and then mapping the log density ratio by different monotonically increasing functions. Compared to previous studies of GANs, our framework is highly theoretically and practically consistent. The limitation of this paper is that our framework does not cover the variants of IPM GANs since these variants give a vector field that is different from the gradient of log density ratios. We will leave it as a future work.

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

## A  PROOFS

### A.1  PROOF OF THEOREM 3.1

**Equilibrium of MonoFlow:** Without loss of generality, we assume the target measure $p$ is properly normalized, i.e., $\int \mathrm{d}p = 1$. To satisfy this assumption, $p$ usually follows a Boltzmann distribution, i.e., $p \propto \exp(-U)$ where the potential energy $U = -\log p$. MonoFlow defines a vector field $v_t = h'\big(\log r_t\big)\big(\nabla_{\mathbf{x}} \log p - \nabla_{\mathbf{x}} \log q_t\big)$ such that by Theorem 4.6 in (Ambrosio et al., 2008), we can write the continuity equation as

$$\frac{\partial q_t}{\partial t} = \mathrm{div}\big(q_t h'\left(\log r_t\right)\left(\nabla_{\mathbf{x}} \log q_t - \nabla_{\mathbf{x}} \log p\right)\big). \tag{26}$$

At equilibrium (stationary distribution), the probability current is zero,

$$q_t h'\left(\log r_t\right)\left(\nabla_{\mathbf{x}} \log q_t - \nabla_{\mathbf{x}} \log p\right) = 0, \tag{27}$$

This condition is required to reflect the boundary conditions of the continuity equation. Note that the equilibrium state of a general continuity equation does not necessarily indicate that the current (flux) has to be zero. MonoFlow is a special case where the drift force of the system is generated by the potential energy (conservative force), such that achieving equilibrium is equivalent to the current being zero.

Since $h'\left(\log r_t\right) > 0$, we directly have $q_t\left(\nabla_{\mathbf{x}} \log q_t - \nabla_{\mathbf{x}} \log p\right) = 0$. Hence $q_t = p$ is the solution to the above differential equation, which is the same as finding the equilibrium of the Fokker Planck equation in Eq. (3).

**The dissipation rate:** For any curve $\{q_t\}_{t\geq 0}$ evolving according to the vector field $\{v_t\}_{t\geq 0}$ in Wasserstein space, the dissipation rate of the functional $\frac{\partial \mathcal{F}(q_t)}{\partial t} = \mathbb{E}_{q_t}\langle \nabla_{W_2}\mathcal{F}(q_t), v_t\rangle$, where the Wasserstein gradient of KL divergence is $\nabla_{W_2}\mathcal{F}(q_t) = \nabla_{\mathbf{x}} \log \frac{q_t}{p}$.

Therefore, the dissipation rate of the KL divergence under MonoFlow is

$$\frac{\partial \mathcal{F}(q_t)}{\partial t} = \mathbb{E}_{q_t}\left[-h'\big(\log r_t\big)\left\|\nabla_{\mathbf{x}} \log \frac{q_t}{p}\right\|^2\right] \leq 0, \tag{28}$$

and equality is achieved if and only if $q_t = p$. Hence, MonoFlow always decreases the KL divergence with time.

### A.2  PROOF OF THEOREM 3.2

Define the functional $\mathcal{F}(q)$ of $f$-divergences as

$$\mathcal{F}(q) = \mathcal{D}_f(p||q) = \int f\left(\frac{p}{q}\right)(\mathbf{x})q(\mathbf{x})\mathrm{d}\mathbf{x}. \tag{29}$$

where $f\colon \mathbb{R}^+ \to \mathbb{R}$ is a convex function and we may further assume that $f$ is twice differentiable.

Let $\phi \in \mathcal{P}(\mathbb{R}^n)$ be an arbitrary test function, the first variation (functional derivative) $\frac{\delta \mathcal{F}}{\delta q}$ is defined as

$$\begin{aligned}
\int \frac{\delta \mathcal{F}}{\delta q}(\mathbf{x})\phi(\mathbf{x})\mathrm{d}\mathbf{x} &= \lim_{\epsilon \to 0} \frac{\mathcal{F}(q + \epsilon\phi) - \mathcal{F}(q)}{\epsilon} \\
&= \frac{d}{d\epsilon}\mathcal{F}(q + \epsilon\phi)\Big|_{\epsilon=0} \\
&= \frac{d}{d\epsilon}\int f\left(\frac{p}{q + \epsilon\phi}\right)(\mathbf{x})\big(q(\mathbf{x}) + \epsilon\phi(\mathbf{x})\big)\mathrm{d}\mathbf{x}\Big|_{\epsilon=0} \\
&= \int \left\{f\left(\frac{p}{q + \epsilon\phi}\right)(\mathbf{x})\phi(\mathbf{x}) - f'\left(\frac{p}{q + \epsilon\phi}\right)(\mathbf{x})\frac{p(\mathbf{x})\phi(\mathbf{x})}{q(\mathbf{x}) + \epsilon\phi(\mathbf{x})}\right\}\mathrm{d}\mathbf{x}\Big|_{\epsilon=0} \\
&= \int \left\{f\left(\frac{p}{q}\right) - f'\left(\frac{p}{q}\right)\frac{p}{q}\right\}(\mathbf{x})\phi(\mathbf{x})\mathrm{d}\mathbf{x}.
\end{aligned} \tag{30}$$

Thus,

$$\frac{\delta \mathcal{F}}{\delta q} = f(r) - r f'(r), \quad \text{where} \quad r = \frac{p}{q}. \tag{31}$$

Recall that the Wasserstein gradient of $\mathcal{F}(q)$ is the Euclidean gradient of the first variation, we have

$$\nabla_{W_2} \mathcal{F}(q) = \nabla_{\mathbf{x}} \frac{\delta \mathcal{F}}{\delta q} = -r f''(r) \nabla_{\mathbf{x}} r. \tag{32}$$

The corresponding vector field is given by the negative Euclidean gradient, see Section 2, therefore the particle flow ODE of $f$-divergences can be written as

$$\mathrm{d}\mathbf{x} = -\nabla_{\mathbf{x}} \frac{\delta \mathcal{F}}{\delta q}(\mathbf{x})\mathrm{d}t = r(\mathbf{x})f''(r(\mathbf{x}))\nabla_{\mathbf{x}} r(\mathbf{x})\mathrm{d}t = r(\mathbf{x})^2 f''(r(\mathbf{x}))\nabla_{\mathbf{x}} \log r(\mathbf{x})\mathrm{d}t. \tag{33}$$

We have $h'(\log r) = r^2 f''(r) > 0$ since $f$ is a convex function, concluding the proof.

### A.3 PROOF OF LEMMA 3.3

This proof is adapted from Lemma 1 and 2 by Moustakides & Basioti (2019). Given the optimization problem

$$\max_{d \in \mathcal{H}} \mathbb{E}_{\mathbf{x} \sim p}\left[\phi\big(d(\mathbf{x})\big)\right] + \mathbb{E}_{\mathbf{x} \sim q}\left[\psi\big(d(\mathbf{x})\big)\right], \tag{34}$$

where $\mathcal{H}$ is a class of measurable functions. We rewrite it as

$$\begin{aligned} \max_{d \in \mathcal{H}} \mathbb{E}_{\mathbf{x} \sim q}&\left[\frac{p(\mathbf{x})}{q(\mathbf{x})}\phi\big(d(\mathbf{x})\big) + \psi\big(d(\mathbf{x})\big)\right] \\ &= \mathbb{E}_{\mathbf{x} \sim q}\left[\max_{d \in \mathcal{H}}\left\{\frac{p(\mathbf{x})}{q(\mathbf{x})}\phi\big(d(\mathbf{x})\big) + \psi\big(d(\mathbf{x})\big)\right\}\right], \end{aligned} \tag{35}$$

we apply the interchange of maximum and integral because the integral operator is independent of $d$. Since the maximum is holding for every fixed $\mathbf{x}$, thus we let the derivative $\frac{\partial}{\partial d(\mathbf{x})}\left[\frac{p(\mathbf{x})}{q(\mathbf{x})}\phi\big(d(\mathbf{x})\big) + \psi\big(d(\mathbf{x})\big)\right] = 0$, we have the optimal $d^*$, the abbreviation of $d^*(\mathbf{x})$, to satisfy

$$r\phi'(d^*) + \psi'(d^*) = 0, \quad r(\mathbf{x}) = \frac{p(\mathbf{x})}{q(\mathbf{x})} > 0 \tag{36}$$

Furthermore, we need to discuss under what sufficient conditions, $d^*$ is the unique maximizer for the above problem. Denote $l(d) = r\phi(d) + \psi(d)$, in order to ensure that $d^*$ is the unique maximizer, $l'(d)$ should satisfy

$$l'(d) > 0, \forall d < d^* \text{ and } l'(d) < 0, \forall d > d^*. \tag{37}$$

We summarize two **sufficient conditions** as:

1. $\phi, \psi$ are concave functions and the resulting mapping $\mathcal{T}(d) := -\frac{\psi'(d)}{\phi'(d)}$ is a bijection.

2. $\phi' > 0$ and the resulting mapping $\mathcal{T}$ is a strictly increasing mapping (also a bijection).

For condition 1, it is obvious $d^* = \mathcal{T}^{-1}(r)$ is the root of Eq. (36) and since $\phi$ and $\psi$ are concave, hence $l(d)$ is concave which satisfies Eq. (37). Therefore, $d^*$ is the unique maximizer.

For condition 2, we can write $l'(d) = [\mathcal{T}(d^*) - \mathcal{T}(d)]\phi'(d)$. Since $\mathcal{T}$ is a strictly increasing mapping and $d^*$ is the maximizer, we have $\mathcal{T}(d^*) - \mathcal{T}(d) > 0$ for $d < d^*$ and $\mathcal{T}(d^*) - \mathcal{T}(d) < 0$ for $d > d^*$. Hence $l'(d)$ satisfies the condition stated in Eq. (37).

In Table 2, $b$-gan satisfies condition 2 and the rest of divergence GANs satisfy condition 1.

Some examples:

- For binary classification, $\phi(d) = \log \sigma(d)$ and $\psi(d) = \log(1 - \sigma(d))$, $r(\mathbf{x}) = \exp(d^*(\mathbf{x}))$.

- Fenchel-duality, $\phi(d) = d$, $\psi(d) = -\tilde{f}(d)$, $r(\mathbf{x}) = \tilde{f}'\big(d^*(\mathbf{x})\big)$ where the convex conjugate is $\tilde{f}(d) = \sup_{r \in \mathrm{dom}f}\{rd - f(r)\}$

- For least-square GAN, $\phi(d) = -(d-1)^2$, $\psi(d) = -d^2$, $r(\mathbf{x}) = \frac{d^*(\mathbf{x})}{1-d^*(\mathbf{x})}$

## B EXPERIMENTS

### B.1 DETAILS FOR 4.3

Table 3: Explicit forms of $f$ and $h$

| | $f(r)$ | $h(u), u = \log r$ |
|---|---|---|
| KL | $-\log r$ | $u$ |
| Forward KL | $r \log r$ | $-u \exp(u)$ |
| Chi-Square | $(r-1)^2$ | $-(\exp(u)-1)^2$ |
| Hellinger | $(\sqrt{r}-1)^2$ | $-(\sqrt{\exp(u)}-1)^2$ |
| Jensen-Shannon (GAN) | $r \log \frac{2r}{1+r} + \log \frac{2}{1+r}$ | $-\exp(u) \log \frac{2\exp(u)}{1+\exp(u)} - \log \frac{2}{1+\exp(u)}$ |
| Exp | $-\exp(1.5 \log r)$ | $\exp(1.5u)$ |

The generator is initialized at:

$$N \left[ \begin{pmatrix} 1.0 \\ 1.0 \end{pmatrix}, \begin{pmatrix} 1.00 & 0.00 \\ 0.00 & 1.00 \end{pmatrix} \right]$$

and the target distribution is

$$N \left[ \begin{pmatrix} 0.0 \\ 0.0 \end{pmatrix}, \begin{pmatrix} 1.00 & 0.80 \\ 0.80 & 0.89 \end{pmatrix} \right]$$

The visualisation results follow the order: KL, Forward KL, Chi-Square, Hellinger, Jensen-Shannon (GAN), Exp. $r_{\text{GAN}}(\mathbf{x})$ uses a simple 2-layer discriminator with Leaky ReLU activation that has logit output as the log density ratio.

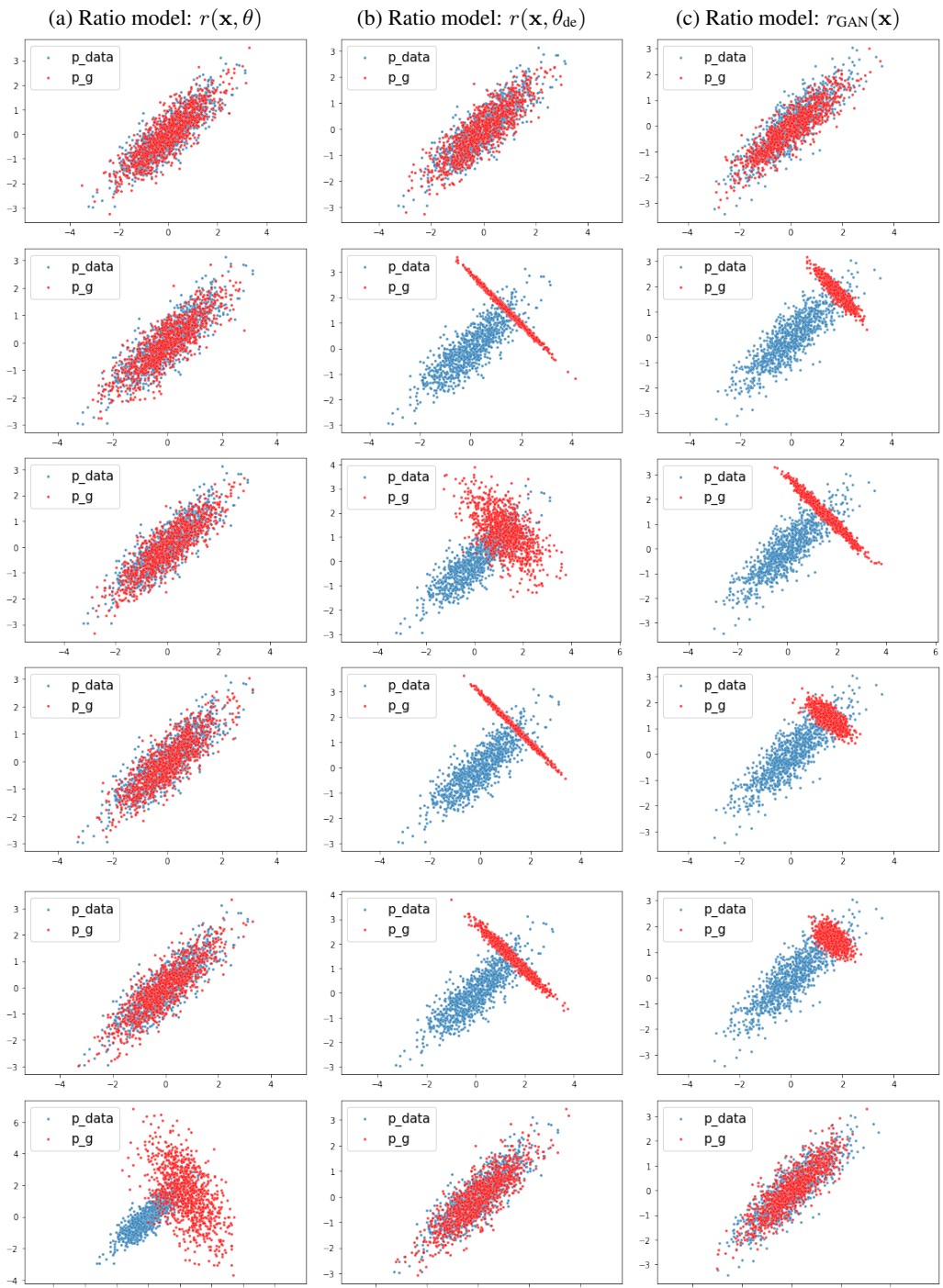

## B.2 Experiment on Vanilla GAN with An Arbitrary Increasing Function

We change the generator loss to: $-\mathbb{E}_{\mathbf{z} \sim p_{\mathbf{z}}} \left[ \arcsinh \left( \left( d(g(\mathbf{z})) - 3 \right)^3 \right) \right]$. This has no meaning for reconstructing any probability divergences, but GAN still works. We did an experiment on Cifar10 and Celeb-A using DCGAN architecture (Radford et al., 2015). The generated samples are shown in Figure 6.

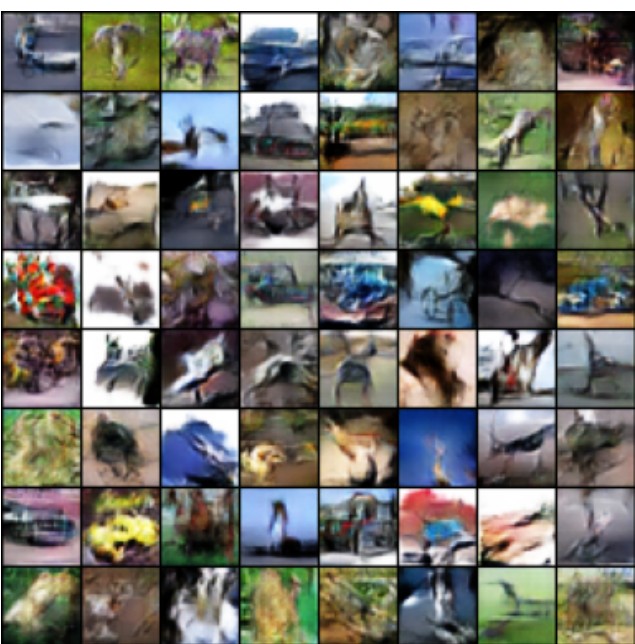

(a) Generated images of Cifar10

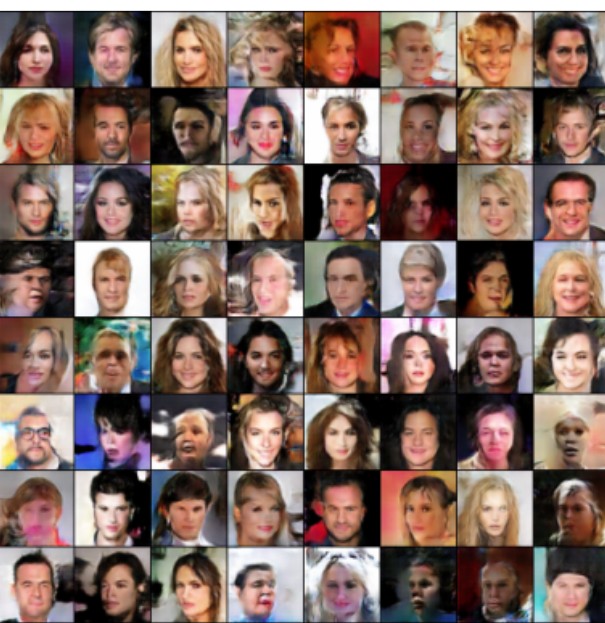

(b) Generated images of Celeb-A

Figure 6: Generated samples via an arbitrary increasing function.

