# OpenReview forum: "MonoFlow: A Unified Generative Modeling Framework for GAN Variants"
_ICLR.cc/2023/Conference — Submitted to ICLR 2023_

### Official Review · Reviewer_4htw · 2022-10-25

**Confidence:** 4
**Correctness:** 2
**Technical Novelty And Significance:** 3
**Empirical Novelty And Significance:** 2
**Recommendation:** 3

**Clarity, Quality, Novelty And Reproducibility:**

### Clarity

While the paper is **mostly clear**, there are **some issues** described above that should be fixed before publication.

### Quality

As previously discussed in this review, **proofs of theoretical results lack details and are some of these results are questionable**. This issue needs to be addressed for this paper to be publishable.

Additionally, the paper should be proofread; cf. this list of typos and formatting problems that should be corrected.

Typos:
 - Abstract (p. 1): "These analysis" -> These analyses / this analysis.
 - Introduction, (p. 1): "Brock et al., 201" -> Brock et al., 2018.
 - Introduction, (p. 2): "insteand" -> instead.
 - Section 3.2 (p. 4): "ascend" -> ascent.
 - Section 3.2 (p. 5): "will lost" -> will lose.
 - Section 3.4 (p. 6): there is a double period at the end of the section.
 - Section 4.1 (p. 6): "can also alternatively minimizes" -> can also alternatively minimize.
 - Section 4.2 (p. 7): there is a double space between "but minimizing it w.r.t." and "$\theta$".
 - Table 2 (p. 7): "means not converges" -> "means it does not converge", "the convergences" -> convergence.
 - Section 4.3, Remark (p. 7): the first quote should be reversed.
 - Section 5.1 (p. 8): "Vainilla" -> Vanilla.
 - Section 6 (p. 9): "Diffussion" -> Diffusion, "denisty" -> density
 - References (p. 10, 11 and 12): capital letters are missing for nome proper nouns.

Formatting:
 - Use `\colon` for function definitions (e.g. $\mathcal{F}$ on p. 2).
 - The differential operator (e.g. in $\mathrm{d}x$) should be upright and not italicized.
 - There should be no space before a footnote mark (e.g. with footnote 2 on p. 2).
 - There should be a period after the abbreviation "Eq" (e.g. on p.2, in Section 3.1).
 - Tables should be properly formatted with the `booktabs` package and ideally placed on top of pages.
 - Section 6: citations in the flow of the sentence should not be parenthesized.
 - Section B: please add a visual way to differentiate generated and data points besides the color for better readability.

### Novelty

As explained above, the proposed MonoFlow is **novel** and could be significant for future research on GAN optimization. However, this novelty should be properly contextualized w.r.t. to previous work as this is not the first paper evidencing these issues in previous theoretical frameworks, or linking GANs and gradient flows.

### Reproducibility

No reproducibility statement is included in the submission. As indicated before in this review, proofs are too informal. There is no code and few details associated to the experimental results. Therefore, **the presented results are not reproducible enough**.

**Strength And Weaknesses:**

### Strengths

The proposed framework is **well motivated** in the paper, as it answers several key issues with previous interpretations of GANs highlighted in the introduction. Treating GANs as gradient flows like in MonoFlow **does solve recurring problems in GAN theory and is a relevant approach**. MonoFlow is **well presented** with a sufficient level of details for non-experts of gradient flows to be able to follow the paper. **Experiments illustrating the validity of the theory** are a nice addition supporting its relevance.

To the best of my knowledge, the key contribution of this paper is **the link between many GAN models and the gradient flow of the reverse KL**. This has **many interesting consequences**:
 - it links conflicting views of GANs and score-based generative models (Section 2);
 - it formalizes why non-saturating losses can make GANs converge, which was a known fact but only understood intuitively;
 - it gives general criteria to design GAN models that converge;
 - it better fits GAN practice than many previous theoretical approaches.

This makes the contributions **novel and potentially significant** for future studies.

### Weaknesses

Unfortunately, this submission is diminished by several important problems.

First of all, I think that there are **issues in the presentation and proofs of theoretical results**. The **formulation of Lemma 3.3 is somewhat convoluted** and should be reformulated. More importantly, the proofs in the appendix of Theorem 3.1, 3.2 and Lemma 3.3 are **too informal** and lack necessary details: I suggest the authors to write complete proofs. Jointly with a **lack of rigorously formulated hypotheses**, this **casts doubt on the presented results**, which is prohibitive. In particular, I found the following problems which the authors should clarify; note that this is not exhaustive and the whole proofs should be rigorously rewritten.
 - Both the main paper and the appendix **lack hypotheses on the optimization space of functions and on manipulated objects**. For example, in Lemma 3.3, in which precise space is $d$ optimized, and what are the hypotheses on $\psi$ and $\phi$?
 - Could the authors explain why $\mathcal{T}$ must be a bijection in Section A.3?
 - I do not understand how the proof of Theorem 3.1 shows that $q_t \to p$. Indeed, as explained in Section A.1, the only equilibrium point is $q_t = p$, but nothing indicates that the gradient flow tends to this equilibrium.

Secondly, the novel contributions of this paper, highlighted above, are **not clearly contextualized w.r.t. existing literature**. While the related work section mentions previous work linking GANs and gradient flows, other works have tackled this topic as well: e.g., Huang & Zhang (2022) and Franceschi et al. (2022, cited elsewhere in the paper). This should be apparent from the very beginning of the paper, either in the introduction or by moving the related work section right after the introduction: while the presented framework and results are novel, they should be contextualized w.r.t. prior work on GANs and gradient flows. Moreover, the motivation for the framework (points 1, 2, 3 in Section 1) and the explanation behind the differences between adversarial training and VDM (Section 4.2) **share similar arguments as Metz et al. (2017) and Franceschi et al. (2022)** regarding gradient computations: this should be further discussed in the paper as well. Finally, I would suggest the authors to amend the qualification of Monoflow as a "unified generative modeling framework", as it is restricted to some GANs, even through covering many models.

Thirdly, there are **some clarity issues** that make it difficult to investigate the correctness and significance of the approach, which are listed below.
 - In Section 1, authors state that "[the discriminator] is a function only depending on samples $x$ and does not include any density information from the generator’s distribution". This statement, that is repeated throuhout the paper, is quite obscure and should be clarified. This is especially the case in Section 4.2 where it helps understanding the differences between adversarial training and VDM. Moreover, I believe that the correct statement would be that this dependency does exist, but it is ignored when optimizing the generator (as expressed by the stop gradient operator of Section 4.3).
 - The contribution of Section 4.1 w.r.t. results of the previous sections are not immediate and should be explicated.
 - Could the authors clarify the differences between convergence requirements for $r(x, \theta)$/$r(x, \theta_{\mathrm{de}})$ and $r_{\mathrm{GAN}}(x)$?
 - The empirical illustration of Section 5.1 does not quite match the theory without explanations. If all considered losses are monotonically increasing functions of the log density ratio, how could $q_t \not\to p$ like in Theorem 3.1? Note that this is related to the raised doubts on this convergence result above in this review.
 - Less importantly, contrary to the statement in Section 3.2 that the discriminator is optimized by a "one-step gradient update", it is also common practice to perform multiple-step optimization of the discriminator between generator updates.

Metz et al. Unrolled Generative Adversarial Networks. ICML 2017.\
Huang & Zhang. GANs as Gradient Flows that Converge. arXiv, 2022.

**Summary Of The Paper:**

This paper introduces a theoretical framework for GANs, called MonoFlow, that challenges their usual understanding as an adversarial minimization of a distance/divergence. It interprets GAN training as the generator following a reparameterized gradient flow of the reverse KL, defined through the log density ratio estimated by the discriminator, yielding convergence results. This more accurate framework encapsulates numerous GAN models without gradient penalties and takes into account common GAN practice like non-saturating losses and discarded gradients for the generator. The validity of the developed theory is then empirically assessed on toy data and simple image datasets.

**Summary Of The Review:**

The proposed framework is well motivated, relevant, interesting and novel. I believe that it could have a significant impact on our understanding of GAN optimization. However, I think that this paper is not ready for publication, because of three main issues: lack of rigor in the theoretical part, improper related work discussion and unclear developments complicating the assessment of the quality of the paper. Therefore, I would recommend to reject this submission and encourage the authors to strengthen it as it could become a great paper for a future top-tier conference such as ICLR.

I am willing to change my evaluation, should any of my concerns or misunderstanding be addressed during the discussion period with the authors or the other reviewers.

---

> ### Author Response · Authors · 2022-11-16
> **Thank you for your feedback**
>
>
> Thanks for acknowledging the novelty and the significance of our paper. The below are the discussions answering your questions. We have fixed typos and addressed major formatting issues. We also added these discussions to our updated version that will be online soon.
>
> ### Q1. Issues in the presentation and proofs of theoretical results
>
> >**Which precise space is $d$ optimized, and what are the hypotheses on $\phi$ and $\psi$?**
>
> There is no restriction for the space of $d$, it can be any measurable functions,
> e.g, a linear model or a neural net. We followed the conventions of standard divergence GANs where $d$'s space is not particularly specified.
>
> For $\phi$ and $\psi$, we added the hypotheses that ensure $d^\ast(x)$ is the unique maximizer. (Direct results from Lemma 2 by Moustakides (2019)).
> $\phi$ and $\psi$ have to satisfy either a or b:
>
> **a**. $\phi$, $\psi$ are concave functions and the resulting mapping $\mathcal{T}(d) : = -\frac{\psi'(d)}{\phi' ( d) }$ is a bijection; **b**. $\phi' >0 $ and the resulting mapping $\mathcal{T}$ is a strictly increasing mapping (also a bijection)..
>
> >**Why $\mathcal{T}$ must be a bijection?**
>
> As we have explained in Lemma 3.3, for the optimal discriminator to be $d^\ast(x) = \mathcal{T}^{-1}[r(x)]$, $\mathcal{T}$ must be a bijection such that its inverse exists.
>
>
>
> >**I do not understand how the proof of Theorem 3.1**
>
> Thanks for pointing it out.
> First, consider the Fokker Planck equation in Eq (3),
>
> $$
> \frac{\partial q\_t}{\partial t} = \text{div}(q\_t(\nabla_x \log q_t - \nabla_x \log p))
> $$
>
> this is the probability evoluation of Langevin dynamics or the log density ratio ODE of Eq(4) where $h$ is an identity mapping.
> The equilibrium is achieved if the probabilty current (flux) $q_t(\nabla_x \log q_t - \nabla_x \log p)$ is zero,
> meaning there is no net flow and $\frac{\partial q_t}{\partial t} = 0 $. Solving the differential equation $q_t(\nabla_x \log q_t - \nabla_x \log p)=0$,
> we have $q_t=p$ if we assume $p$ is a proper normalized distribution. This convergence is a well established theory, see section 1.2 and 1.3 by https://userswww.pd.infn.it/~orlandin/fisica_sis_comp/fokker_planck.pdf. There is a sufficient boundary condition where the target density $p$, e.g. a Boltzmann distribution,  decays fast at $x \to \infty$ to satisfy the normalization condition (Eq 1.85 in https://userswww.pd.infn.it/~orlandin/fisica_sis_comp/fokker_planck.pdf). In order to satisfy the boundary condtion, probability current must be uniformly zero everywhere at equilibrium.
>
> **1. The equilibrium of MonoFlow**:
> For MonoFlow, the continuity equation is
>
> $$
> \frac{\partial q_t}{\partial t} = \text{div} (q_t h'(\log \frac{p}{q_t})(\nabla_x \log q_t - \nabla_x \log p))
> $$
>
> if we let the probablity current (net flow) be zero, we have
>
> $$
> q_t h'(\log \frac{p}{q_t})(\nabla_x \log q_t - \nabla_x \log p) =0
> $$
>
> Since $h' > 0$, we have $q_t (\nabla_x \log q_t - \nabla_x \log p) =0$, this is the same situation as we have for the Fokker Planck equation.
> Hence $q_t \to p$.
> **We added the hypothesis in the proof, assuming the target $p$ is a Boltzmann distribution, a general assumption in energy-based models, with the potential energy $-\log p$**.
>
> **2. The dissipation rate**:
>
> $$
> \frac{\partial\mathcal{F}(q_t)}{\partial t} = \mathbb{E}\_{q_t}{\langle \nabla\_{W_2} {\mathcal{F}(q_t)}, v_t \rangle}
> $$
>
> is a standard property of continuity equation and Wasserstein gradient flow, see Chap 11 by Ambrosio et al (2008). Replace $v_t$ by the vector filed of MonoFlow, it is trivial to see the rate wrt. KL divergence is always negative if $q_t \neq p$. Since the dissipation is the time derivative of a functional, that means this functional is always decreasing with time evolution.
>
> Given the above, theorem 1 tells us that MonoFlow is always decreasing KL while it converges to $p$.
>
> **A helpful inituitive understanding of the convergence of MonoFlow**.
> Since $h'$ is postive scalar function, it can be folded into the step size of Euler discritization, see Eq (12). Different scales of $h'$ are reflected on the evolution speed of simulating the log density ratio ODE in Eq (4) which is the associated ODE of Langevin dynamics.

---

> > ### Author Response · Authors · 2022-11-16
> > **Answers for the following questions**
> >
> > ### Q2. the novel contributions of this paper, highlighted above, are not clearly contextualized w.r.t. existing literature
> > >**While the related work section mentions previous work linking GANs and gradient flows, other works have tackled this topic as well: e.g., Huang & Zhang (2022) and Franceschi et al. (2022, cited elsewhere in the paper).**
> >
> > Thanks for reminding us of the work by Huang & Zhang (2022). Huang & Zhang (2022) is a special case of Gao et al (2019) where they choose the $f$-divergence to be Jensen-Shannon divergence but they did not cite Gao et al (2019) in their paper. Eq (2.14) in Huang & Zhang (2022) satisfies our **theorem 3.2** where $f$ is the function following JSD. However, as it is dicussed in our paper, the gradient flow of GANs is not related to JSD, neither under the vanilla loss or the non-saturated loss as their associated $h$ does not follow a specific type of $f$-divergences. Gao et al (2019) was cited in our paper under **theorem 3.2** and we added more explainations to discuss the difference. Our **theorem 3.2** also indicates MonoFlow is more general than existing Wasserstein gradient flows of $f$-divergences, as we only need to design an increasing function $h$ where the resulting continuity equation does not need to follow the form of gradient flows in the Wasserstein space.
> >
> > Franceschi et al (2022) focuses on the NTK view given a genenral generator losses and they also discussed some IPM GAN losses, while our paper focuses on how a specific generator loss can work for divergences GANs. We added the comparison to Franceschi et al (2022).
> >
> >
> > >**the explanation behind the differences between adversarial training and VDM (Section 4.2) share similar arguments as Metz et al. (2017) and Franceschi et al. (2022)**
> >
> > Metz et al. (2017) mentioned the optimal discriminator is a smooth function of $p_g$ but this smoothness gaurantees are lost during the practical algorithm. Franceschi et al. (2022) also claims that "the dependency of the optimal discriminator on the generator’s parameters are discarded". Their statements are different ways of experssing the stopping gradient operator which is used in our paper, but they did not further explain how this issue results in the difference between VDM and adversarial training. We have added citations of these two papers in section 4.2 for the discussion of VDM.
> >
> > >**Finally, I would suggest the authors amend the qualification of Monoflow as a "unified generative modeling framework", as it is restricted to some GANs, even though covering many models.**
> >
> > We added "divergence GANs" in the title.

---

> > > ### Author Response · Authors · 2022-11-16
> > > **Answers for the following questions**
> > >
> > > ### Q3. Clarity issues
> > >
> > > >**In Section 1, the authors state that "[the discriminator] is a function only depending on samples  and does not include any density information from the generator’s distribution". This statement, that is repeated throuhout the paper, is quite obscure and should be clarified. This is especially the case in Section 4.2 where it helps understanding the differences between adversarial training and VDM. Moreover, I believe that the correct statement would be that this dependency does exist, but it is ignored when optimizing the generator (as expressed by the stop gradient operator of Section 4.3).**
> > >
> > > Yes, your understanding is correct. It might depend on how we interpret the word "dependency". To avoid ambiguity, we change the words to "the discriminator cannot capture the variability of $p_g$".
> > >
> > > >**The contribution of Section 4.1 w.r.t. results of the previous sections are not immediate and should be explicated.**
> > >
> > > This section explains why the intuitive way of maximizing the lower bound of a divergence and then minimizing this divergence can lead to the convergence of $p_g$ (as mentioned in the previous question, the variability of $p_g$ is lost for the discriminator), because the objectives for generators are increasing functions. As this is the standard adversarial game used by some GAN variants such as Goodfellow (2014) and Nowozin (2016), we further discussed how this differs from VDM. We added more explanations to improve the connections to the previous sections.
> > >
> > > >**Could the authors clarify the differences between convergence requirements for $r(x , \theta)$,  $r(x , \theta\_{de})$ and $r\_{GAN}(x)$?**
> > >
> > > For $r(x , \theta)$, in order to obtain the convergence, the cost function should be $f(r)$ where $f$ is a convex function. This is derived by viewing the divergence as a function of $p_g$ or its parameter $\theta$, such that the minimum is achieved via applying Jensen-Shannon inequality - the reason why we need $f$ to be convex. The $f$-divergence is a cost function of $\theta$ where $x$ is integrated out,
> > >
> > > $$
> > > \mathcal{D}_f(p_g) = \text{Cost}(\theta)=\int f(r(x, \theta)) p_g(x;\theta)dx \geq f(\int r(x, \theta)p_g(x;\theta)dx) = f(1)=0,
> > > $$
> > >
> > > The minimum is achieved when $r$ is constant which means $p_g=p\_{data}$. This is a standard variational inference problem.
> > >
> > > For $r(x , \theta\_{de})$ and $r\_{GAN}(x)$, the requirement is the objective should be $-h(\log r)$ where $h$ is an increasing function. In this case, we can never write the divergence as a cost function of $\theta$. This follows our framework as amortizing MonoFlow into a parametric generator.
> > >
> > > >**The empirical illustration of Section 5.1 does not quite match the theory without explanations.**
> > >
> > > Our Section 5.1 **does match** the theory. We analyzed how the derivative $h'$ influences the convergences of GANs by Eq (6). For vanilla loss and MLE loss, the derivatives are nearly zeros when $d(x)$ (log density ratio) is less than zero. From Eq (6), we can see $\mathrm{d}x_t \approx 0$ because $h'\approx 0$. This means a slow convergence and this is the main reason causing gradient vanishing **(because the rescaled vector field is too small)**, just like multiplying an extremely small learning rate. In section 5.2, we have shown that the gradient vanishing problem can be simply fixed by just shifting the function to get non-zero $h'$.
> > >
> > > >**Less importantly, contrary to the statement in Section 3.2 that the discriminator is optimized by a "one-step gradient update".**
> > >
> > > This is true, we can apply multiple-step optimization to get a more accurate approximation of density ratios. We have changed it to "a small number of gradient updates" in our paper.
> > >
> > >
> > > ### Q4. Reproducibility
> > > We provide the code in the https://anonymous.4open.science/r/MonoFlow-7C32
> > >
> > > Reference
> > > 1. Metz et al. Unrolled Generative Adversarial Networks. ICLR 2017.
> > > 2. Huang & Zhang. GANs as Gradient Flows that Converge. arXiv, 2022.
> > > 3. Gao et al. Deep generative learning via variational gradient flow. ICML, 2019.
> > > 4. Moustakides et all. Training neural networks for likelihood/density ratio estimation. arXiv, (2019).
> > > 5. Nowozin et al. f-gan: Training generative neural samplers using variational divergence minimization." NeurIPS (2016).
> > > 6. Franceschi et al. A neural tangent kernel perspective of gans. ICML (2022).

---

> > ### Comment · Reviewer_4htw · 2022-11-20
> > **Welcome Clarifications but not Publishable As Is**
> >
> > I would like to thank the authors for providing necessary clarifications of their theoretical results and additional improvements w.r.t. related work and clarity. I carefully read their response as well as the other reviews and responses. However, I still have reservations on the validity and presentation of the claims of theoretical results which, I believe, **necessitate another round of reviewing** to be carefully checked, as explained in the following. Therefore, I have to maintain my initial "reject" recommendation.
> >  - **Convergence of $q_t$ to $p$.** I do agree that the KL between $q_t$ and $p$ is decreasing throughout training and that the equilibrium, under reasonable hypotheses, is $q_t = p$. However, I still have doubts on the convergence $q_t \to p$, as I did not find any reference of such property under the authors' assumptions in the provided references. Note that this does not mean that this result is incorrect, but this claim requires a clear reference, especially as convergence under these hypotheses is not shown empirically in the paper (which could be caused by either extremely slow convergence as the authors claim, or non-convergence as well).
> > - **Quality of proofs & results presentation.** I appreciate the better quality of the proofs, but they remain somewhat rushed and unclear. For example, the inversion of the expectation and max operators in Eq. (35) does not hold for arbitrary classes of measurable functions $\mathcal{H}$ as claimed in the corresponding proof and section; this should be corrected. More generally, the paper still suffers from not clearly stating hypotheses in the main results, which makes proofs hard to follow and assess as assumptions are only introduced in the course of the proofs. This leads to misleading results like Lemma 3.3 suggesting that $\mathcal{T}$ being a bijection is a result, whereas it is, to the best of my understanding, an assumption in Section A.3.
> >
> > As I said in my review, this paper does raise interesting and novel points that help us to better understanding GAN optimization by tying it with the KL gradient flow. I truly regret recommending a rejection for an innovative work, but questionable claims and proofs harm the publishability of the paper. With a rigorous rewriting to fix these issues, I believe it could be accepted at the next round of submission in a top-tier conference.
> >
> > I look forward to discussing with the other reviewers about this submission. I remain open for any clarification or discussion on possible misunderstandings.
> >
> > Less importantly, I would have some remarks on the authors' response which clarified many of my other initial concerns.
> >  - The Boltzmann distribution assumption for $p$ might not fit well the case of GANs. It may be better to explicitly state the normalization condition mentioned by the authors and discuss why most data distributions satisfy it in practice.
> >  - I believe that Franceschi et al. (2022) do make a difference between VDM and adversarial training beyond IPMs, as they explain like in this article that this gradient flow interpretation changes the loss optimized by the generator. Nonetheless, they fail to find this loss for non-IPM GANs in their setting. I would see both articles as complementary.
> > - Regarding the convergence requirements in Table 2, I am not sure to understand the authors' response. Do they mean that convergence for $r(x, \theta)$ is assessed theoretically? Or are all convergence checks obtained qualitatively from the visualization of Appendix B.1?

---

> > > ### Author Response · Authors · 2022-11-23
> > > **Thank you for your reply (2/2)**
> > >
> > > Less importantly,
> > >
> > > >**The Boltzmann distribution assumption might not fit well in the case of GANs**
> > >
> > > This assumption is widely used in generative modeling, as it follows the form $\exp(-f(x))$ where $f(x)$ is the potential, e.g., it can be an output of neural networks. Many milestone algorithms rely on this assumption. For example,
> > > - Boltzmann Machines (Hinton, 2007).
> > > - All classes of energy models (Song, 2021).
> > > - The earliest discretized version of score-based diffusion models (Song, 2019), because Langevin dynamics converges to Boltzmann distributions.
> > > - Even GAN itself was explained as an energy model (Che, 2020).
> > >
> > > >**Regarding the convergence requirements in Table 2, I am not sure to understand the authors' response. Do they mean that convergence is assessed theoretically? Or are all convergence checks obtained qualitatively from the visualization of Appendix B.1?**
> > >
> > > The convergence requirements have theoretical guarantees, as for the first model, $f$ has to be convex and for the later two, $h$ has to be increasing. We provide an empirical study to verify the theoretical guarantees. The experimental result of convergence is monitored by comparing the values of $\theta$ to the true value, but for conciseness, we did not list them in the paper and only add a description in Table 2 to state how we assess the convergence. The visualization is included as a complement to support the assessment. The code is also included in the anonymous link for reproducibility.
> > >
> > >
> > > References:
> > > 1. Nguyen, XuanLong, Martin J. Wainwright, and Michael I. Jordan. "Estimating divergence functionals and the likelihood ratio by convex risk minimization." IEEE Transactions on Information Theory 56.11 (2010): 5847-5861.
> > > 2. Geoffrey E. Hinton. "Boltzmann Machines", https://www.cs.toronto.edu/~hinton/csc321/readings/boltz321.pdf. 2007
> > > 3. Song, Yang, and Diederik P. Kingma. "How to train your energy-based models." arXiv preprint arXiv:2101.03288 (2021).
> > > 4. Song, Yang, and Stefano Ermon. "Generative modeling by estimating gradients of the data distribution." NeurIPS (2019).
> > > 5. Che, Tong, et al. "Your gan is secretly an energy-based model and you should use discriminator driven latent sampling." NeurIPS (2020).
> > > 6. Nowozin, Sebastian, Botond Cseke, and Ryota Tomioka. "f-gan: Training generative neural samplers using variational divergence minimization." NeurIPS (2016).

---

> > > ### Author Response · Authors · 2022-11-23
> > > **Thank you for your reply (1/2)**
> > >
> > > Dear reviewer,
> > >
> > > Thanks for your reply.
> > >
> > > We answer your questions below. Many of our theoretical results follow the convention of previous GAN papers, especially Lemma 3.3.
> > >
> > > > **Convergence of $q_t$ to p.**
> > >
> > > As mentioned in the previous reply, the proof requires the assumption of Boltzmann distribution equipped with potential energy. **Given this assumption, the sufficient and necessary condition for the continuity equation $\frac{\partial q_t}{\partial t}=\text{div}J$ achieving equilibrium is the probability current $J$ equals zero**
> > >
> > > For the Boltzmann distribution to be normalized, the probability current $J$ vanishes at infinity because there is no mass at infinity,
> > >
> > > $$
> > > J(x) = 0, \text{ if } x \to \infty
> > > $$
> > >
> > > see Eq (1.85) in the [reference.](https://userswww.pd.infn.it/~orlandin/fisica_sis_comp/fokker_planck.pdf)
> > >
> > > Then for 1D case, see Eq (1.92) and Eq (1.93), the probability current (flux) is zero because it is constant and it has to be zero to satisfy the above condition at infinity. [This reference](https://userswww.pd.infn.it/~orlandin/fisica_sis_comp/fokker_planck.pdf) uses reflecting boundary condition within [a, b], the same derivation directly applies to the natural boundary condition by changing $a$ and $b$ to infinity, see https://www2.ph.ed.ac.uk/~dmarendu/ASP/Section16.pdf (Under Eq (3)).
> > >
> > > **We can generalize it to multidimensional cases.**
> > >
> > > At equilibrium, $\text{div} J=0$ (divergence-free) means the left side of
> > > the continuity equation is zero such that the probability $q_t$ does not change with time. And use our assumption where the whole system comes with potential energy (conservation force), so the vector field $J$ is a [conservative vector field](https://en.wikipedia.org/wiki/Conservative_vector_field) such that it is curl-free, i.e., $\nabla \times J=0$.
> > >
> > > Given the above, the current $J$ is both divergence-free and curl-free, hence it satisfies the [Laplace equation](https://en.wikipedia.org/wiki/Laplacian_vector_field)
> > >
> > > $$
> > > \nabla^2 J = 0
> > > $$
> > >
> > > The solution of the Laplace equation is a harmonic function, a harmonic function attains its maximum and minimum at its boundaries (infinity). Since we assume the fast decay boundary condition, so the solution to the Laplacian equation is $J=0$ everywhere because its maximum and minimum are zeros.
> > >
> > > Hence the same as the 1D case, for the continuity equation with potential energy to achieve equilibrium, **the sufficient and necessary condition is the probability current $J$ equals zero.** These are some properties of the equation of Mathematical Physics, we do not discuss these results in detail in the paper, we simply use these main results-- $J=0$ implies the equilibrium. We are happy to add necessary derivations or references if required.
> > >
> > > > **Especially as convergence under these hypotheses is not shown empirically in the paper**
> > >
> > > The empirical result is shown in the section 5.1 and 5.2 as the GAN example and an additional example of using arbitrary increasing function whose derivative deviates from zero is in Appendix B.2.
> > >
> > > >**For example, the inversion of the expectation and max operators in Eq. (35) does not hold for arbitrary classes of measurable functions**
> > >
> > > We **did not use** the terminology "arbitrary classes" in our paper. We used "a class of any measurable functions" or "a class of measurable functions". This might be confusing, perhaps **"a class of all measurable functions"** is better. This is true and a similar proof can be found in a well-known paper of $f$-divergences Nguyen (2010) where the functional class is the set of all measurable functions, see Lemma 1 by Nguyen (2010).
> > >
> > > >**More generally, the paper still suffers from not clearly stating hypotheses in the main results, which makes proofs hard to follow and assess as assumptions are only introduced in the course of the proofs**
> > >
> > > The purpose of Lemma 3.3 is to summarize and reformulate the relationship between the optimal discriminator and the density ratio $\frac{p_{data}}{p_g}$ for divergence GANs, therefore we call it a "lemma" and put additional conditions in the Appendix proof. This relationship can also be found in all divergence GANs papers where they write the optimal discriminator in terms of $p_{data}$ and $p_g$. For example, the proof of the proposition 1 by Goodfellow (2014), is much more informal than our Lemma 3.3, where they directly use the exchangeability of integral and supremum to find the optimal $d$ without indicating any assumptions.
> > >
> > > >**Lemma 3.3 suggesting that $\mathcal{T}$ being a bijection is a result, whereas it is, to the best of my understanding, an assumption in Section A.3.**
> > >
> > > This is an assumption or restriction that has to be satisfied in our statement of Lemma 3. "Must be a bijection" means we have to ensure it is bijection for the availability of inverse.  We are open to changing it to other better words to make it more clear as an assumption.

---

> > > > ### Comment · Reviewer_4htw · 2022-11-24
> > > > **Remaining doubts**
> > > >
> > > > First of all, I would like to thank you for this discussion!
> > > >
> > > > I feel that the issues we have been debating about result from unclear and/or ambiguous assertions and assumptions in the paper causing confusion and misunderstandings. This is a major problem, especially for a theory-oriented paper, that requires in my opinion partially rewriting it so that the claims and results can be properly assessed, with clearly stated assumptions. This demands significant changes that, I believe, can only be addressed with a new version of the paper.
> > > >
> > > > **Regarding convergence of $q_t$ to $p$.**
> > > > To the best of my understanding, you proved both in the paper and your latest response that:
> > > >  - $\mathcal{F}$ (the KL) is strictly decreasing as long as $q_t \neq p$;
> > > >  - if the system achieves equilibrium, then $q_t = p$.
> > > >
> > > > However, one of the most important claim of the paper is that $q_t$ converges to $p$, which the latter results do not suffice to prove. Therefore, I believe you should clarify both the meaning and the assertion of "convergence to the target distribution" (Section 4, p. 6), as I do not see any proof of this convergence in the paper.
> > > >
> > > > **Regarding the assumptions.**
> > > > I do agree that some assumptions are standard. However, I mainly criticize the lack of clarity in their introduction: they are not clearly highlighted in the main paper and even introduced in the course of the proofs. This forbids a clear and fair assessment of the validity of the claims. It is true that the original article of Goodfellow et al. (2014) was also theoretically questionable, but I would argue that 1. its significant contributions go way beyond its sole theoretical results, 2. standards of publication are nowadays, fortunately, higher, and 3. these approximations also caused some misunderstandings in the community regarding how GANs operate. Hence, I consider the raised assumptions problems in the current submission as important, even though they are not the only reason that I recommend rejection.

---

> > > > > ### Author Response · Authors · 2022-11-24
> > > > > **Regarding the convergence**
> > > > >
> > > > > Dear reviewer,
> > > > >
> > > > > Thank you for your reply. The continuity equation (Fokker Planck equation) has a corresponding Markov process. That is why the equilibrium is called the stationary distribution or the invariant distribution. From that perspective, given any initial distribution, $q_t$ will evolve to the equilibrium. This is also straightforward from physics or statistical mechanics, as any gas system evolves given sufficient time to the thermodynamical equilibrium where the distribution is described by the Maxwell–Boltzmann statistics (Second law of thermodynamics).
> > > > >
> > > > > In order to show $q_t \to p$, we only need to justify if $q_t =p$, the system achieves equilibrium. But this does not indicate how the KL divergence changes with time, so we utilized the dissipation theory which tells us KL decreases while $q_t \to p$.
> > > > >
> > > > > For the reference of Markov process, see the direct derivation http://awibisono.github.io/2016/08/22/continuity-equation.html. Or the Markov semigroup theory in section 1.2 by this draft https://chewisinho.github.io/main.pdf

---

> > > > > > ### Comment · Reviewer_4htw · 2022-11-25
> > > > > > **Precise Reference to Convergence Result**
> > > > > >
> > > > > > I would like to thank you for the references. However, without any specific reference to a given result, I do not see any convergence result in the provided papers that would support the claim that $q_t$ converges to $p$ in the same conditions as in the current submission (i.e. for an arbitrary $h$ with $h' > 0$).
> > > > > >
> > > > > > To clarify this point, could you please:
> > > > > >  1. explicitly and mathematically state your claim of convergence (there may be many different and non-equivalent definitions of $q_t \to p$)?
> > > > > >  2. give the precise reference to a convergence result that would be applicable in the setting of the current submission?

---

> > > > > > > ### Author Response · Authors · 2022-11-26
> > > > > > > **Precise Reference**
> > > > > > >
> > > > > > > The first [reference](http://awibisono.github.io/2016/08/22/continuity-equation.html) reformulates the content from the book by Michael C. Mackey (1992)
> > > > > > >
> > > > > > > In Mackey (1992), equations (4.11) to (4.14) describe the relationship between continuity-equations and the Frobenius-Perron operator (Markov operator). (4.15) states that the necessary and sufficient condition for stationary distribution (for Markov process) is that the system achieves equilibrium. It is obvious that $q_t=p$ is the unique solution for the equilibrium.
> > > > > > >
> > > > > > >
> > > > > > > This convergence of a Markov process is usually a weak convergence, $\lim_{t\to \infty} \int f dq_t =  \int f dp $. We do not specify the convergence with advanced math languages here. In our paper, we say "as $t \to \infty$, $q_t$ evolves to p". This should not cause any ambiguity since it is commonly used in stochastic processes just like we describe the marginal evolution of a simple Markov chain. We will add the weak convergence to the revision of our paper.
> > > > > > >
> > > > > > > Reference:
> > > > > > > Mackey, Michael C. Time's arrow: The origins of thermodynamic behavior. Courier Corporation, 1992.

---

> > > > > > > > ### Comment · Reviewer_4htw · 2022-11-26
> > > > > > > > **Interesting New Elements for Future Version**
> > > > > > > >
> > > > > > > > I would like to thank you for the reference. These new convergence considerations are interesting and would greatly improve the paper if included in a future version with sufficient details in the proof.

---

### Official Review · Reviewer_Wf4w · 2022-10-26

**Confidence:** 2
**Clarity, Quality, Novelty And Reproducibility:** 1. The idea is novel and well present…
**Correctness:** 3
**Technical Novelty And Significance:** 4
**Empirical Novelty And Significance:** 4
**Recommendation:** 6

**Strength And Weaknesses:**

Strengths:
1. An interesting idea with nice theories and consistent practical findings. First, it is very interesting that from the perspective of the gradient flow in this paper, all log density ratio based GAN variants are methodologically equal. These GAN variants all are different methods of estimating the bijection of the log density ratio and then mapping the log density ratio by different monotonically increasing functions. Second, the analysis of generator loss, especially Fig. 2, to reveal the failure of vanilla and MLE loss and the success of NS and logit loss from the perspective of the adopted monotonically increasing mapping, really provides a novel and practical insight.
2. The simple trick to fix vanilla GAN works well and refreshes me. The trick is also practical support to their theories.
3. The idea is well presented in this paper.

A potential improvement:
An algorithmic inspiration for the generator loss has been proposed. I am wondering if such a framework can provide inspiration for the discriminator loss to compare different estimations of the bijection of the log density ratio, such as JSD divergence and f-divergence. The authors should discuss this.

Some typos:
1. In the contributions: “insteand” → “instead”
2. Under Eq. (21), the two $\tilde{f}(d*)$ are defined differently.

A further thought:
1. In the related work, the authors somehow unify GANs and diffusion models from particle evolution. I wonder if such a framework can provide some views to discuss the advantages/disadvantages of the vector field of MonoFlow and the vector field by time-dependent neural networks of diffusion models.


**Summary Of The Paper:**

This paper provides new insight to understand GANs based on log density ratios as a gradient flow in the Wasserstein space. Specifically, the bi-level step of adversarial training in GANs is regarded as it first estimates the vector field of the gradient flow (by discriminator), and next the generator is updated to learn to draw particles guided by the vector field. In particular, the particle evolution can be rescaled via an arbitrary monotonically increasing mapping. Such a new insight enables an analysis of what types of generator loss functions can lead to the success of training GAN and identifies the difference between VDM and adversarial training.

**Summary Of The Review:**

I think this is a very interesting paper. The perspective of Wasserstein gradient flow indeed brings out a novel and practical insight to the GANs community, especially for the design of the generator loss. I particularly appreciate the analysis of the practical effectiveness of generator loss. However, I am not an expert on the theory of GAN, so it is possible that I did not identify the mistakes in the theoretical results.

---

### Official Review · Reviewer_wjv3 · 2022-10-29

**Confidence:** 4
**Correctness:** 3
**Technical Novelty And Significance:** 3
**Empirical Novelty And Significance:** 3
**Recommendation:** 6

**Clarity, Quality, Novelty And Reproducibility:**

The paper can improve in clarity, particularly in laying out the background for readers less exposed to PDEs and continuity equations. The quality and novel are good in my opinion, and the experiments are sufficiently explained to be reproducible.

**Typo fixes**

Page 2: “and instead we should treat GANs”

Page 5: “density ratio estimation will lose density information”

**Strength And Weaknesses:**

**Strengths**

The proposed theory exposes an interesting connection between Wasserstein gradient flow and divergence GANs, and can better explain why certain choices of final activations (leading to different divergences) work. This new theory is closer to practice in my opinion, and as such can be very valuable in better understanding divergence GANs and how to improve them.

**Weaknesses**

The paper does not read well in my opinion, some parts can use more clear explanation and better connections to background theory, in particular the convergence of Euclidean ODE to Wasserstein PDE (convergence of the empirical measure to q) deserves more attention.

Another issue is not discussing some limitations when connecting to practice, in particular, the choice of discriminator itself (and its regularizers such as R1 R2 etc) is very consequential in practice, while the paper does not discuss this choice under the lens of its theory. The new theory assumes the ratio is estimated accurately, but this is of course not true in practice, and as a consequence the updates are not strictly following the vector field of the Wasserstein gradient flow in practice, it is important to discuss to what extent will such inconsistencies affect the theory.

Another issue is that of the three disconnects you raise about original GAN theory in page 2, I don't agree with 1 and 2. For 1, the divergence's first term derivative is uniformly zero wrt generator so how does not optimizing it matter? For 2, I'm not sure I understand your argument here, can you elaborate on this given Proposition 2 of original Goodfellow et al. theory?

Finally, I think the title and abstract of the paper should mention “divergence GANs”, rather than GANs, to emphasize that IPM GANs are not covered by the proposed theory.

**Summary Of The Paper:**

This paper introduces a new formulation for adversarial training as a special case of wasserstein gradient flow. The discriminator update is explained as approximating the vector field of a gradient flow in the Wasserstein space, and the generator update as moving particles (samples) in this vector field. This perspective unifies several GAN variants, and provides a unified insight as to why these variants work. The paper also provides some toy experiments to verify its predictions about the reasons why some variants might work or not.

**Summary Of The Review:**

This paper proposes a new theory of divergence GAN that is closer to practice and can better explain the dynamics of their training and what can make/break them. Background must be discussed more substantially, and some missing connections to practice need to be clarified and discussed. Overall, I think this paper can be valuable for both understanding and improving divergence GANs.

### Update post-rebuttal
I thank the authors for their clarifications and answering my concerns. I consider the paper interesting and valuable, however, the paper is still lacking in clarity and very hard to follow (skips some underlying assumptions and discussions to better place its results wrt existing theory). Given that the paper's main contribution is theoretical, clarity and mathematical rigor is a must, and so I maintain my borderline score of 6.

---

> ### Author Response · Authors · 2022-11-16
> **Thank you for your feedback**
>
> Thanks for your insightful comments. We answer your questions as below.
>
>
> ### Q1 Explanations and better connections to background theory.
>
> Thanks for pointing it out. Our framework MonoFlow started at the probability flow ODE (log density ratio ODE) of Langevin dynamics which is well studied. Our theorem 3.2 relies on the Wasserstein gradient flow theories to derive the first variation of $f$-divergence to design $h$. We added some connections linking them to the background theory in our paper and it will be online soon.
>
> ### Q2 The concerns about Arguement 1 and 2 in the Introduction.
> In this paper, we only want to point out the disconnections between the common practice of GAN training and the theoretical framework (Goodfellow et al (2014)), as it is well-known that there already exists an inconsistency which is the non-saturated trick. The main reason of this disconnection is that adversarial games require $p_g$ as a functional variable ( $p_g$ is expressed by a parametric neural network $G$). In proposition 1 by Goodfellow et al (2014), the optimal discriminator is a function of $G$ and $x$ satisfies
>
> $$
> D^\\ast(G, x) = \\frac{p\_{data}(x)}{p\_{data}(x)+p_g(x)}=\sigma[d^\ast(G, x)]
> $$
> where the small $d^\ast(G, x) = \log \frac{p_{data}(x)}{p_g(x)}$ is the logit output in our paper. Given this optimal $D^\ast(G, x)$, the Jensen-Shannon divergence (JSD) is expressed by (see Eq (4) or Eq (5) by Goodfellow et al (2014)),
>
> $$
> \begin{align*}
> C(G) &= \mathbb{E}\_{x\sim p\_{data}}\left[ \log D^\ast(G, x) \right] + \mathbb{E}\_{x\sim p\_g}\left[ \log (1-D^\ast(G, x)) \right] \\\\
> &= \mathbb{E}\_{x\sim p\_{data}}\left[ \log \frac{p_{data}(x)}{p_{data}(x)+ p_g(x)} \right] + \mathbb{E}\_{x\sim p\_g}\left[ \log \frac{p\_g(x)}{p\_{data}(x)+ p_g(x)} \right] \\\\
> & = \Big[KL(p\_{data}||\frac{p\_{data}+p_g}{2}) - \log(2)\Big] + \Big[KL(p\_g||\frac{p\_{data}+p_g}{2}) - \log(2)\Big]
> \end{align*}
> $$
>
>
> we can see that the **first term of JSD is dependent on $G$** and this JSD $C(G)$ is a function of $G$. However, in the practical algorithm, $D$ is trained only with samples drawn from two fixed distributions, therefore the optimal discriminator should be $D^\ast(G_{\text{detach}}, x)$ which cannot capture the variability of $p_g$ and this is the reason why we drop out the first term of JSD in practice. This issues was also mentioned by Metz et al. (2017): "the optimal discriminator is a smooth function of $p_g$ but this smoothness gaurantees are lost during the practical algorithm" and Franceschi et al. (2022): "the dependency of the optimal discriminator on the generator’s parameters are discarded".
>
> >**Reviewer question: I don't agree with 1 and 2. For 1, the divergence's first term derivative is uniformly zero wrt generator so how does not optimizing it matter?**
>
> As we can see, lossing this dependence caused that the generator loss of vanilla GAN actually approximates the second term,
>
> $$
> KL(p\_{g}||\frac{p\_{data}+p_g}{2}) - \log(2)=\mathbb{E}\_{x\sim p\_g}\left[ \log(1-D^\ast(G_{\text{detach}}, x)) \right]
> $$
>
> this is not JSD **(our argument 1)** but a KL divergence up to a constant.
>
> >**Reviewer question: For 2, I'm not sure I understand your argument here, can you elaborate on this given Proposition 2 of original Goodfellow et al. theory?**
>
> Proposition 2 by Goodfellow et al. (2014) requires that given the optimal discriminator, we can write the problem of minimizing JSD as
>
> $$
> \min_{p\_g} JSD(p\_{data}||p\_{g}) \Leftrightarrow \min_G C(G)
> $$
>
> However, this should not be viewed as a divergence minimization problem **(our argument 2)** beacause the smoothness of the optimal discriminator on $p_g$ is lost. In the practical algorithm, the divergence is recovered from a trained discriminator $D(x)$, this discriminator does not contain any density information from $p_g$.  In a variational divergence minimization problem, we need $D^*(G, x)$ to construct a probability divergence such that this divergence is a smooth function of $p_g$ as we explained it in section 4.2.
>
> **Under our framework:**
>
> The vanilla loss and non-saturated loss work because they just construct a monotonically increasing function $h$ and the derivative $h'$ rescales the vector field of MonoFlow (vanilla loss can work in toy data sets if two distributions are close such that the estimated log ratio is close to zero where $h'$ deviates from zero). In Appendix B.2, we have also shown that GAN can work with an arbitrarily increasing function $h$ as long as its derivative deviates from zero when $d(x)$ is less than zero.

---

> > ### Author Response · Authors · 2022-11-16
> > **Answers for the following two questions**
> >
> > ### Q3 The concern about the choice of discriminators and the accuracy of density ratio estimation.
> >
> > Yes, our framework requires the assumption of accurate ratio estimations. For density ratio estimation, we follow the same argument by the conventions of divergence GANs, e.g., the logit output $d^*(G,x)$ is the log density ratio is a direct result of **Proposition 1** by Goodfellow et. al. (2014) as we explained in the previous question. How to obtain an accurate density ratio estimator is still an open problem and this might be a future work of MonoFlow, like discussing other types of divergence GANs.
> >
> > Density ratio estimation via binary classifications is stable because the cross entropy is associated with Sigmoid activation where extreme values are bounded such that overflow risks in the computation are low. We also found gradient penalties are essential to avoid discriminator outputting 'nan' values in training GANs if the density ratio model is trained via the Donsker-Varadhan representation or the f-divergence representation of KL divergence, see Nowozin (2016) or Belghazi (2018). But this is not discussed in our paper, we focus on the aspect of generator losses and we want to correct the misunderstanding of GANs as a divergence minimization problem. It is known that divergence estimation is equivalent to density ratio estimations, we refer to Sugiyama (2012) for more details on density ratio estimation.
> >
> > ### Q4 The ambiguity of the title.
> >
> > Thanks for pointing it out. We added "divergence GANs" in the title.
> >
> > References
> > 1. Goodfellow  et al. "Generative adversarial networks." NeurIPS (2014).
> > 2. Belghazi et al. "Mine: mutual information neural estimation." ICML (2018).
> > 3. Nowozin el al. "f-gan: Training generative neural samplers using variational divergence minimization." NeurIPS (2016).
> > 4. Sugiyama  Masashi, Taiji Suzuki, and Takafumi Kanamori. Density ratio estimation in machine learning. Cambridge University Press (2012).
> > 5. Metz et al. Unrolled Generative Adversarial Networks. ICLR 2017.
> > 6. Franceschi et al. A neural tangent kernel perspective of gans. ICML (2022).

---

### Decision · Program_Chairs · 2023-01-20

**Decision:**

Reject

**Justification For Why Not Higher Score:**

Pls find the three limitations mentioned above

**Justification For Why Not Lower Score:**

N/A

**Metareview: Summary, Strengths And Weaknesses:**

This paper reveals a new insight into the GAN community, based on log density ratios as a gradient flow in the Wasserstein space. Specifically in GAN, its discriminator update is explained as approximating the vector field of a gradient flow in the Wasserstein space, while its generator update as moving particles  in this vector field. In this way, the new thinking integrates several GAN variants, and provides a unified insight.

In experiments, the paper uses some toy data to verify its predictions.

While it is a interesting paper, it needs significant improvements to clarify and strengthen its claims as it is not rigorous in its arguments.




**Summary Of Ac-Reviewer Meeting:**

1. Lemma 3.3. should include two sufficient conditions in A.3 proof to make it precise. In addition, it relates to results in previously published papers, which should be clearly clarified.

2. Though the analysis for the failed convergence of vanilla GAN is interesting and reasonable, it is not directly related to the assumption of mono increasing function but the gradient of h being not too small. This somehow violates the results of Theorem 3.1. The vanilla GAN has a mono increasing function $h$ and thus is supposed to converge according to the theorem.

3. Regarding the convergence proof of Theorem 3.1, a reference to support the convergence $KL->0$ when $t->\infinity$ should be included.